



# Energy and mass exchange at an urban site in mountainous terrain – the Alpine city of Innsbruck

Helen Claire Ward[1], Mathias Walter Rotach[1], Alexander Gohm[1], Martin Graus[1], Thomas Karl[1], Maren Haid[1], Lukas Umek[1], Thomas Muschinski[1]

[1]Department of Atmospheric and Cryospheric Sciences, University of Innsbruck, Innsbruck, Austria

*Correspondence to:* Helen C. Ward (helen.ward@uibk.ac.at)





**Abstract**
This study represents the first detailed analysis of multi-year near-surface turbulence observations for an urban
area located in highly complex terrain. Using four years of eddy covariance measurements over the Alpine city of
Innsbruck, Austria, the effects of the urban surface, orographic setting and mountain weather on energy and mass
exchange are investigated. In terms of surface controls, findings for Innsbruck are in accordance with previous
studies at city-centre sites. The available energy is partitioned mainly into net storage heat flux and sensible heat
flux (each comprising about 40% of the net radiation, $Q^*$, during summer daytimes). The latent heat flux is small
by comparison (only about 10% of $Q^*$) due to the small amount of vegetation present but increases for short
periods (6-12 h) following rainfall. Additional energy supplied by anthropogenic activities and heat released from
the large thermal mass of the urban surface helps to support positive sensible heat fluxes in the city all year round.
Annual observed $CO_2$ fluxes (5.1 kg C $m^{-2}$ $y^{-1}$) correspond well to both modelled emissions and expectations based
on findings at other sites with a similar proportion of vegetation. The net $CO_2$ exchange is dominated by
anthropogenic emissions from traffic in summer and building heating in winter. In contrast to previous urban
observational studies, the effect of the orography is examined here. Innsbruck's location in a steep-sided valley
results in marked diurnal and seasonal patterns in flow conditions. A typical valley-wind circulation is observed
(in the absence of strong synoptic forcing) with moderate up-valley winds during daytime, weaker down-valley
winds at night (and in winter) and near-zero wind speeds around the times of the twice-daily wind reversal. Due
to Innsbruck's location north of the main Alpine crest, south foehn events frequently have a marked effect on
temperature, wind speed, turbulence and pollutant concentration. Warm, dry foehn air advected over the surface
can lead to negative sensible heat fluxes both inside and outside the city. Increased wind speeds and intense mixing
during foehn (turbulent kinetic energy often exceeds 5 $m^2$ $s^{-2}$) help to ventilate the city, illustrated here by low $CO_2$
mixing ratio. Radiative exchange is also affected by the orography, for example incoming shortwave radiation is
blocked by the terrain at low solar elevation. Interpretation of the dataset is complicated by distinct temporal
patterns in flow conditions and the combined influences of the urban environment, terrain and atmospheric
conditions. The analysis presented here reveals how Innsbruck's mountainous setting impacts the near-surface
conditions in multiple ways, highlighting the similarities with previous studies in much flatter terrain and
examining the differences, in order to begin to understand interactions between urban and orographic processes.
**1    Introduction**
Driven by the need to better understand the environment in which we live, the number of micrometeorological
studies in urban areas has grown considerably over the last twenty years, expanding the temporal coverage, breadth
of surface and climatic conditions and variety of locations observed. Urban eddy covariance measurements have
been made across a range of surface types, including urban parks (e.g. Kordowski and Kuttler, 2010; Lee et al.,
2021), vegetated suburban neighbourhoods (e.g. Grimmond and Oke, 1995; Crawford et al., 2011; Ward et al.,
2013), densely-built city centres (e.g. Grimmond et al., 2004; Gioli et al., 2012; Kotthaus and Grimmond, 2014a)
and high-rise districts (Ao et al., 2016). A few studies have investigated different sites within the same city, for
example in Basel (Rotach et al., 2005), Melbourne (Coutts et al., 2007b) and Łódź (Offerle et al., 2006), in order
to isolate the effect of surface characteristics on exchange processes under similar synoptic and climatic conditions.
While the majority of studies focus on mid-latitude European or North American cities, observations have also
been made in Asia (e.g. Moriwaki and Kanda, 2004; Liu et al., 2012), Africa (Offerle et al., 2005b; Frey et al.,
2011) and South America (e.g. Crawford et al., 2016), at higher latitudes (Vesala et al., 2008) and in (sub-) tropical
climates (e.g. Weissert et al., 2016; Roth et al., 2017). However, very few studies have addressed surface exchange
and turbulence characteristics for urban areas in hilly or mountainous regions.
For historical reasons many cities are situated in complex topography such as in valleys or basins or along
coastlines (Fernando, 2010). With high densities of people living in such areas, knowledge of how cities and their
surroundings interact is of great relevance to the health and well-being of the human population. Compared to non-
urban, flat, horizontally homogeneous terrain both urban and mountainous environments can have major effects
on atmospheric transport and exchange. Both environments present physical obstacles to the flow and are
characterised by extreme spatial variability. Orography gives rise to various phenomena which interact across a
range of spatial and temporal scales, including the mountain-plain circulation, valley winds, slope winds, gap
flows, downslope windstorms, mountain waves and cold-air pools (Whiteman, 2000). In the urban environment
anthropogenic activities release heat and pollutants, the large thermal mass of buildings and manmade surfaces





store and release a significant amount of heat, and the lack of vegetation and pervious surfaces limits water
availability, all of which impact surface-atmosphere exchange and boundary layer characteristics (Oke et al.,
60 2017).

Cities in mountainous terrain often experience extreme weather such as windstorms, heavy snowfall and flooding.
Air quality can be a major issue, especially in urbanised valleys during winter when inversions and terrain can
prevent dispersion of pollutants (e.g. Velasco et al., 2007; Largeron and Staquet, 2016). On the other hand, slope
and gap flows can transport pollutants into adjacent valleys or across long distances (Gohm et al., 2009; Fernando,
2010). Local- and mesoscale flows affect the city climate and can act to ameliorate or exacerbate heat stress (e.g.
a cool sea breeze versus warm foehn) (Hirsch et al., 2021). There is a real need for turbulence observations to
develop process understanding, evaluate model performance and improve predictive capabilities in complex
terrain, particularly when meteorologically based tools and expertise are used to inform planning or policy
decisions that have direct consequences for human and environmental health (Rotach et al., submitted). Measures
that have been successfully applied to other cities may have inadvertent effects when applied to a different city,
especially one with very different surroundings. For example, attempts to mitigate the urban heat island can
interfere with the circulation patterns in complex terrain and have a detrimental effect on air quality (Henao et al.,
2020). Numerical modelling is indispensable for investigating such effects, but if models are applied to areas where
they have not been carefully evaluated, the output may be inaccurate and measures could be implemented that
have unintended consequences or even act to exacerbate rather than ameliorate the situation.
Most previous urban-related studies in or near complex terrain focused on dispersion of pollutants (e.g. Allwine
et al., 2002; Doran et al., 2002; Velasco et al., 2007) or used routinely measured variables such as air temperature
and near-surface wind speed to demonstrate the presence of an urban heat island and/or regional circulations (e.g.
Miao et al., 2009; Giovannini et al., 2014). More recently, Doppler-wind lidars have been used to capture flow
patterns in urbanised valleys, such as above the cities of Passy (Sabatier et al., 2018), Stuttgart (Adler et al., 2020)
and Innsbruck (Haid et al., 2020). The scarcity of turbulence observations in complex terrain, especially urban
complex terrain, means there is very little information available on how the orographic setting of a city affects
surface-atmosphere exchange. In Salt Lake Valley the relation between cold-air pools and $CO_2$ mixing ratio has
been studied in winter (Pataki et al., 2005) and the impact of land cover differences on $CO_2$ fluxes has been
examined in summer (Ramamurthy and Pardyjak, 2011). A short campaign in suburban Christchurch indicated
differences in energy partitioning between foehn flow and sea breeze conditions (Spronken-Smith, 2002) and a
summertime campaign in Marseille found small differences between katabatic flow and sea breeze conditions
(Grimmond et al., 2004).
The focus of this study is the city of Innsbruck, Austria, located in a steep-sided Alpine valley. Innsbruck thus
represents an urban site in extremely complex terrain, in contrast to most previous studies where the terrain is
typically at a greater distance from the site and/or much less complex (mostly flat). The two main research goals
are to investigate how surface-atmosphere exchange of energy and mass for a city in a complex orographic setting
compares to other sites in the literature which are in much less complex terrain, and to examine the effect of the
orographic setting on near-surface conditions in the city. The multi-year dataset analysed here allows for
characterisation of the radiation budget, energy balance terms and carbon dioxide exchange, exploration of
temporal variability from sub-daily to interannual timescales and investigation of a variety of conditions. The paper
is organised as follows. Section 2 provides details of the site, instrumentation and data processing. In Section 3 the
source area characteristics are explored and in Section 4 an overview of the climate and meteorological conditions
is given to set the dataset in context. Sections 5-9 comprise the presentation and discussion of results, including
flow and stability (Section 5), radiation and energy balance (Sections 6-7), $CO_2$ fluxes (Section 8) and the effects
of different flow regimes (Section 9). Findings are summarised and conclusions drawn in Section 10. A second
paper (Ward et al., in prep.) will examine the turbulence characteristics in more detail.

## 2 Methods

### 2.1 Site description

Innsbruck is a small city in the northern European Alps with a population of 132,000 (Statistik Austria, 2018). The
city is built along the east-west oriented Inn Valley and extends approximately 7 km in the along-valley direction
and 2-3 km in the cross-valley direction (Figure 1). Most of the built-up area is confined to the reasonably flat



valley floor. The northern edge of the city is bounded by the Nordkette mountain range, which rises steeply from
the valley floor; the southern side of the valley is less steep. Agricultural land and smaller urban settlements lie to
the west and east of the city and the valley slopes are mainly forested (up to the tree line). The valley floor is about
570 m above sea level (a.s.l.) and the surrounding terrain rises to over 2500 m a.s.l. with a peak-to-peak distance
across the valley of 15-20 km. The north-south oriented Wipp Valley exits into the Inn Valley just to the south of
the city.
Turbulence observations have been made on top of the university building close to the centre of Innsbruck since
2014. In May 2017 the measurement tower was relocated from the north-eastern side to the south-eastern corner
of the university building rooftop as part of the development of the Innsbruck Atmospheric Observatory (IAO).
The IAO comprises a suite of instruments for studying urban climate and air quality (Karl et al., 2020). Besides
basic meteorological variables and in situ fast-response measurements of wind, temperature, water vapour and
carbon dioxide, many trace gases and aerosols are also observed (e.g. Karl et al., 2017; Deventer et al., 2018; Karl
et al., 2018), plus vertical profiles of wind using a Doppler lidar (Haid et al., 2020; Haid et al., 2021) and
temperature and humidity using a microwave radiometer (Rotach et al., 2017). The focus of this study is on the
surface-atmosphere exchange of momentum, heat and mass, observed using the eddy covariance (EC) technique.
Innsbruck has a small historical core surrounded by predominantly residential areas and industrial zones towards
the edges of the city. In the city centre the buildings are closely packed and typically around 6 storeys; away from
the centre the buildings are more spread out and slightly lower (3-4 storeys). Based on the local climate zones of
Stewart and Oke (2012), the majority of the city is 'open midrise' with 'compact midrise' in the old city core.
The IAO site is located a few hundred metres south-west of the city core. The mean building height within a radius
of 500 m is 17.3 m and mean tree height is 10.1 m. The modal building height, $z_H$ is approximately 19 m. The
zero-plane displacement height, $z_d$, is estimated at 13.3 m (based on 0.7 times the modal building height since the
mean building height is reduced by small buildings in courtyards which do not impact the flow (Christen et al.,
2009)) and the roughness length, $z_0$, at 1.6 m (Grimmond and Oke, 1999). The average land cover composition
within 500 m of IAO is 31% buildings, 24% paved surfaces, 18% roads, 19% vegetation and 8% water. The Inn
River flows from south-west to north-east at a distance of about 100-200 m from IAO (Figure 2). A more detailed
source area analysis is presented in Section 3.
**2.2    Instrument details**
A sonic anemometer (CSAT3A, Campbell Scientific) and closed-path infrared gas analyser (EC155, Campbell
Scientific) are installed on a lattice mast at a height of 9.5 m above the rooftop of the university building
(47°15'50.5'' N 11°23'08.5'' E, elevation 574 m a.s.l.), giving a sensor height $z_s$ of 42.8 m above ground level
(i.e. $z_s/z_H = 2.3$). The three wind components, sonic temperature and molar mixing ratios of water vapour and
carbon dioxide are logged at 10 Hz (CR3000, Campbell Scientific). A four-component radiometer (CNR4, Kipp
and Zonen) at a height of 42.8 m provides incoming and outgoing shortwave and longwave radiation, and
temperature and relative humidity are also measured (HC2S3, Campbell Scientific). Rainfall is recorded by a
weighing rain gauge (Pluvio, OTT Hydromet) at 2 m above ground level a few hundred metres south-west of IAO
(47°15'35.5'' N 11°23'03.2'' E).
**2.3    Data processing**
Data are processed to 30-min statistics using EddyPro (v7.0.7, LI-COR Biosciences). The following standard steps
are implemented: despiking of raw data, double rotation to align the wind direction with the mean 30-min flow,
time lag compensation by seeking maximum covariance, correction of sonic temperature for humidity (Schotanus
et al., 1983), and correction for low and high frequency losses (Moncrieff et al., 2004; Fratini et al., 2012).
Subsequent quality control removes data when instruments malfunction and during maintenance, when the wind
direction (WD) is within ±10° of the direction of sonic mounting (309°), when the magnitude of the pitch angle
exceeds 45°, when data fall outside physically reasonable thresholds (absolute limits and a despiking test by
comparing adjacent data points), or when conditions are non-stationary (following Foken and Wichura (1996) with
a threshold of 100%). As for other urban studies, no data were excluded on the basis of skewness or kurtosis tests
and the so-called integral turbulence characteristic tests (Foken and Wichura, 1996) have not been applied here.
These tests are based on typical values and scaling relations observed over simpler surfaces and thus may not be
appropriate for more complex sites (Crawford et al., 2011; Fortuniak et al., 2013; Järvi et al., 2018). Moreover,





the applicability of scaling relations to this dataset is one of the aspects we wish to analyse (Ward et al., in prep.).
Following quality control, 83%, 72% and 79% of sensible heat ($Q_H$), latent heat ($Q_E$) and $CO_2$ ($F_{CO2}$) flux data are
available for the four-year study period: 01 May 2017-30 April 2021. All data are presented in local Central
European Time (CET = UTC+1).

**2.4    Additional measurements**

Data from short-term field campaigns and various monitoring stations are used here to support analysis of the IAO
dataset. As part of the PIANO project investigating foehn winds (Haid et al., 2021; Muschinski et al., 2021; Umek
et al., 2021), EC measurements were made at a height of 2.5 m at a grassland site at Innsbruck airport 3.4 km west
of IAO (47°15'19.4'' N 11°20'34.2'' E, 579 m a.s.l., Figure 1). The station, hereafter referred to as FLUG, was
operated from 15 September 2017-22 May 2018 (although data transmission issues resulted in a low data capture
rate for September). A similar closed-path eddy covariance system (CSAT3A + EC155) and four-component
radiometer (CNR4) to those at IAO were deployed, along with two soil heat flux plates at 0.05 m depth (HFP01,
Hukseflux), a temperature and humidity probe (HC2S3), a tipping bucket rain gauge (ARG100, Campbell
Scientific) and soil temperature sensors (107, Campbell Scientific). Data were logged at 20 Hz and processed in
the same way as for the IAO station (Section 2.3). The soil heat flux at the surface was estimated from the average
heat flux measured by the plates adjusted to account for the heat stored in the soil layer between the plate and the
surface based on the soil temperature at 0.02 m depth. This adjustment makes a considerable difference to the
magnitude and phase of the soil heat flux. Comparison of the FLUG dataset with IAO is helpful for distinguishing
urban-related characteristics from other controls and offers some insight into spatial variability in the Inn Valley.
Additional meteorological data from several stations installed as part of the PIANO campaign (labelled P2-4, P5-
8, PAT and THA in Figure 4) and those operated by the Austrian national weather service ZAMG (Z1-3 in Figure
4, S in Figure 1) are also used to investigate spatial variability.

**3    Source area analysis at IAO**

To assist interpretation of the IAO dataset the flux footprint parameterisation of Kljun et al. (2015) was used to
provide an indication of the likely source area characteristics and their variability under different conditions. Figure
2 shows the estimated source area for the study period along with the footprint-weighted land cover composition
as a function of wind direction. The shape of the source area reflects the predominance of along-valley winds
(Section 5). Since the area around the flux tower is fairly homogeneous, the land cover composition of the footprint
does not change considerably with stability or wind direction. There is a slightly larger contribution from
vegetation with increasing stability as the footprint extends further from the tower beyond the city centre. The total
impervious (paved and road) surface fraction varies little with wind direction (at about 40-50%) but the proportion
of roads is greater for easterly winds (≈30%) than westerly winds (≈10%). The eastern sector also has the greatest
proportion of buildings (around 40%) and least vegetation (10%). For the western sector there is slightly more
vegetation (15-20%) and the river comprises up to about 15% of the source area. The aggregated source area
composition for the study period is similar to the average land cover within 500 m (given in Section 2.1), with a
slightly lower fraction of vegetation and slightly higher fractions of buildings and paved surfaces reflecting the
greater weight of the footprint closer to the tower. On average, 70%/80% of the footprint lies within a radius of
500 m/700 m from the tower.

**4    Meteorological conditions during the study period**

Meteorological conditions during the study period are summarised in Figure 3. Innsbruck has a humid continental
climate with cool winters, warm summers and strong seasonality. Average monthly temperatures range from -
0.1 °C in January to 19.8 °C in July and mean annual precipitation is 886 mm (1981-2010 normals for Innsbruck
University (ZAMG, 2021)). Precipitation occurs throughout the year with most rainfall in summer (Figure 3h)
when convective storms are frequent. Snow cover down to the valley floor is common during winter and can last
several weeks at rural locations along the valley (and longer at higher altitudes); in the city snow melts much faster
due to the higher temperatures and it is usually quickly cleared from roads. The increased surface albedo, $\alpha$, during
times of snow cover can be seen clearly in Figure 3a.
The study period 01 May 2017-30 April 2021 was warmer and sunnier than the long-term (1981-2010) average.
Overall 2018 was the warmest year on record in Austria, and 2017, 2018, 2019, 2020 and 2021 were 0.8, 1.9, 1.5



and 1.4 and 0.6 °C warmer than normal in Innsbruck (ZAMG, 2021). April 2018 and June 2019 were particularly
hot and sunny (4.5 and 4.8 °C warmer than normal), whereas September 2017, February 2018 and May 2019 were
much cooler (≥ 2 °C) than normal (with September 2017 and May 2019 also being much cloudier than normal).
While 2017 and 2019 were wetter than normal, 2018 and the first half of 2020 were drier than normal (Figure 3h).
Winter 2017-18 and 2018-19 were particularly snowy. At IAO, the observed daily mean temperature ranged from
a minimum of -9.2 °C in February 2018 to a maximum of 28.1 °C in June 2019 (Figure 3c) and the lowest (highest)
temperature recorded was -13.5 °C (37.7 °C).

## 5    Flow characteristics in and around Innsbruck

### 5.1    Spatiotemporal variability

Flow patterns in mountainous terrain are extremely complex and show a high degree of spatial variability. Flow is
generally channelled along valleys with the dominant wind directions corresponding to the orientation of the valley
axis at a particular point (e.g. compare stations in the Inn Valley with stations in the Wipp Valley in Figure 4a).
On mostly clear-sky days with weak synoptic forcing, a valley-wind circulation often develops (e.g. Zardi and
Whiteman, 2013). These thermally driven mesoscale circulations lead to a twice-daily wind reversal with up-slope
and up-valley flows during the day and down-slope and down-valley flows during the night. Typical thermally
driven circulation patterns have been documented previously in the Inn Valley and surrounding valleys (e.g.
Vergeiner and Dreiseitl, 1987; Lehner et al., 2019). For flat sites on the valley floor (such as IAO or FLUG) flow
tends to be either up- or down-valley, while for sloping sites up- and down-slope winds are also observed (e.g. at
sites PAT, P5 and Z3 in Figure 4a). The up-slope winds usually precede the up-valley winds in the morning and
the down-slope winds precede the down-valley winds in the evening (e.g. easterly down-slope winds at PAT in
Figure 4b).
The strength and timing of the valley-wind circulation depends on meteorological conditions as well as
characteristics of the valley such as its width, height, orientation and surface cover (e.g. Wagner et al., 2015;
Leukauf et al., 2017). The up-valley flow tends to begin and end earlier in the Wipp Valley than in the Inn Valley
around Innsbruck (Dreiseitl et al., 1980). For the October example shown in Figure 4b, the up-valley flow begins
at around 10:00 CET and ends at around 16:30 CET in the Wipp Valley (sites PAT and P3), whereas in the Inn
Valley the up-valley flow begins in the afternoon (12:00-15:00 CET) and continues until the evening (18:00-21:00
CET) – although the timing varies considerably throughout the year (see Figure 5).
At the intersection of two or more valleys the flow field can be especially complex as the individual valley-wind
systems with their different magnitudes and forcings interact. Inflow or outflow from side valleys can affect the
wind field at some distance from the side valley exit. For example, site P6 in central Innsbruck mainly records the
east-west Inn Valley circulation but also detects the southerly katabatic flow from the Wipp Valley seen here as a
change in wind direction from easterly (up-valley flow in the Inn Valley) to southerly (down-valley flow in the
Wipp Valley) once the flow in the Wipp Valley reverses in the afternoon and before the down-valley flow in the
Inn Valley has fully established (Figure 4a-b). However, this outflow from the Wipp Valley is not seen less than
a kilometre away at IAO. Coplanar scans with Doppler wind lidar help to fill in the gaps between point
measurements and give an indication of the extreme spatial variability (Figure 4c, see also Haid et al. (2020)). For
the example shown, strong southerly wind speeds in the Wipp Valley exit jet can be seen mixing with weaker
winds in the Inn Valley. Using Doppler lidars over a larger area (e.g. as in Adler et al. (2020) for the Neckar
Valley, Stuttgart) would be useful for understanding these interacting flows and their role in the distribution of air
pollution within the city.
The long-term wind direction distribution at IAO is roughly bimodal, with more westerly down-valley winds than
easterly up-valley winds (Figure 4d). Although the main wind directions are roughly aligned with the axis of the
Inn Valley (about 75-255° at IAO), winds blowing down-valley are slightly more southerly and winds blowing
up-valley are slightly more northerly. The reason for this 20-30° difference between the valley axis and the main
wind directions is not clear. For simplicity, up-valley winds in the Inn Valley will be referred to here as easterly
(rather than east-north-easterly) and down-valley winds as westerly (rather than south-westerly).
The strong southerly winds visible in Figure 4d are foehn events (warm, dry, downslope windstorms). Due to the
location of the Brenner Pass (the lowest pass in the main Alpine crest) at the top of the Wipp Valley, strong cross-



Alpine pressure gradients can lead to south foehn in the Wipp Valley (about 20% of the time according to Plavcan
et al. (2014)) which frequently reaches Innsbruck in spring and autumn (Mayr et al., 2004).
Because mountain winds are restricted by the terrain and closely connected to thermal and dynamical forcing,
there are strong temporal signatures in the wind regime and associated turbulence. Figure 5 shows the monthly
and diurnal variation in flow, stability and turbulence at IAO for all available data. Although this figure combines
various weather types and flow regimes, on average wind direction at IAO has a clear seasonal and diurnal cycle,
with westerly winds overnight and in the morning, and easterly winds during the afternoon. The duration of the
up-valley flow is greatest during summer, typically beginning late morning (10:00-12:00 CET) and reversing in
the late evening (20:00-22:00 CET), while for winter days there is not always a transition to up-valley flow and,
if one does occur, the up-valley flow lasts only a few hours during the late afternoon and early evening (Figure
5f). Similar patterns have been observed previously in Innsbruck (Vergeiner and Dreiseitl, 1987), in the Adige
Valley, Italy (Giovannini et al., 2017), in the Rhone Valley, Switzerland (Schmid et al., 2020) and in the western
United States (Stewart et al., 2002), for example.
Down-valley wind speeds ($U$) are 0-2 m s$^{-1}$ all year round at IAO (but can be larger aloft) and remain fairly constant
during the night although sometimes show a small maximum in the morning. The up-valley wind speed increases
as the up-valley flow establishes and peaks at around 4 m s$^{-1}$ in the late afternoon a few hours after the peak in
temperature (Figure 5c, e). Around the time of the wind direction transition wind speeds are usually very low.
These near-zero wind speeds in the middle of the day offer little relief from thermal stress in summer. Friction
velocity ($u_*$) is reasonably high due to the rough urban surface ($\geq 0.2$ m s$^{-1}$ even at night, Figure 5b). Turbulent
kinetic energy ($TKE$) follows a similar temporal pattern to wind speed but with a slightly broader peak and larger
values earlier in the day resulting from buoyancy production in the morning (Figure 5a). Dynamic instability
(expressed as the stability parameter $\zeta = (z_s - z_d)/L$, where $L$ is the Obukhov length) is greatest during the morning
hours; in the afternoon the atmosphere becomes more neutral as wind speed increases with the strength of the up-
valley flow. After the peak up-valley wind, conditions usually become more unstable again but, in a few cases,
stable conditions are observed (Figure 5d, h) when the air temperature is close to surface temperature, wind speeds
are low and the sensible heat flux is small and negative.
Stable stratification close to the surface is rarely observed at IAO ($\zeta > 0.1$ only 4.7% of the time). Studies in other
densely built urban areas find similarly rare occurrences of near-surface stable conditions (Christen and Vogt,
2004; Kotthaus and Grimmond, 2014a). Even in cities with cold winters such as Montreal (Bergeron and Strachan,
2010) and Helsinki (Karsisto et al., 2015) the proportion of stable conditions is below about 10%. Within the study
period, no days were identified with persistent near-surface stable stratification at IAO, in contrast to FLUG, which
is usually stable overnight (see Section 7) and experiences several days with stable stratification in winter,
particularly when there is snow cover. Even though strong negative heat fluxes are observed at IAO during foehn
in autumn and winter (Figure 3g, Section 5.3), these are mostly classified as neutral due to the high wind speeds.
Interestingly, the proportion of unstable conditions in the late afternoon is greater in winter than summer (Figure
5h), as no strong up-valley wind develops in winter. Heating of buildings also contributes additional energy to the
urban atmosphere which helps to maintain unstable conditions in winter (Section 7.1). Note that although the near-
surface atmosphere in the city is almost always unstable or neutral, the valley atmosphere is often stably stratified
higher up (based on temperature profiles from radiosonde and microwave radiometer, data not shown) and, in
winter, strong temperature inversions can reduce vertical mixing and contribute to poor air quality. Further
research is needed concerning the three-dimensional structure of the mountain boundary layer (Lehner and Rotach,
2018), particularly the transition between near-surface conditions and the valley atmosphere aloft.
Although several overall trends emerge from the average of this multi-year dataset, when looking at each day
individually there is a great deal of variability resulting from the complex urban environment and, more
significantly, its orographic setting. Observed wind direction is often highly variable, particularly when the wind
speed is low. Some days have easterly flow in the early morning or lasting for several days, which may be low-
level cold-air advection from the Alpine foreland. On some days an up-valley wind establishes but is then
interrupted and sometimes later resumes and sometimes not. In some cases this can be related to a drop in solar
radiation, rainfall, foehn or outflow from convection. Although easterly winds occur in the afternoon and evening
on most days, textbook valley-wind days at IAO are surprisingly rare (as has also been reported for the Inn Valley
by Vergeiner and Dreiseitl (1987) and Lehner et al. (2019)).





### 5.2    Valley-wind case study

To examine valley-wind features more closely, observations at IAO and FLUG from four clear-sky days in April 2018 are shown in Figure 6. A fairly typical valley-wind circulation is seen in both the Inn Valley and the Wipp Valley on 18, 19 and 21 April. In both valleys, wind speed is greatest for the well-established up-valley flow and minima at the times of flow reversal can be seen more clearly than in the averages in Figure 5c. For these clear-sky conditions, the diurnal course of net radiation and temperature is smooth (radiation data at FLUG in the early mornings is missing because of dew on the sensor). At IAO, surface temperature ($T_{sfc}$) is much larger than air temperature ($T_{air}$) during the morning, which drives large $Q_H$; in the afternoon and evening $T_{sfc}$ remains comparable to $T_{air}$ and $Q_H$ is smaller but remains mostly positive. At FLUG, $T_{sfc}$ is smaller than in the city and slightly larger than $T_{air}$ during the morning but falls rapidly in the afternoon and drops below $T_{air}$ well before sunset. Therefore, $Q_H$ peaks in the morning and becomes negative in the afternoon, supplying energy to maintain large $Q_E$ until sunset.

At first, many of the characteristics appear similar on 20 April, but the lack of a wind reversal, high wind speeds and the slightly distorted diurnal cycle of potential temperature at Steinach (S in Figure 1) point to foehn in the Wipp Valley (Figure 6b, h-i) which seems to reach IAO and FLUG for a very short period in the afternoon. This example highlights the difficulty of distinguishing between different flow regimes in certain cases, and it is quite typical to have days with a valley-wind type circulation that are also influenced by foehn or other dynamically forced flows. At IAO and FLUG, the matching of potential temperature to that in the Wipp Valley for a short period and the slightly larger *TKE* values, coupled with the southerly (rather than easterly) wind at IAO suggest the foehn reached these stations for a few hours in the afternoon.

### 5.3    Characteristics and effects of foehn

South foehn is most common in Innsbruck in spring (with an average of 22 days during March-May) and autumn (with an average of 13 days during September-November), although there is strong interannual variability (Mayr et al., 2004). The higher frequency of southerly foehn winds in spring and autumn can be seen in the wind direction distributions in Figure 5f-g. In spring, foehn often reaches the floor of the Inn Valley sometimes bringing several days of continuous or almost continuous foehn, strong southerly winds and intense mixing in and around Innsbruck, while foehn events tend to be shorter in autumn. Depending on the local strength and depth of the cold pool in the Inn Valley, a breakthrough may occur on one side of the city but not the other (Haid et al., 2020; Muschinski et al., 2021; Umek et al., 2021). Foehn is rare in summer as the synoptic situation is unfavourable. In winter the cold pool in the Inn Valley often prevents a breakthrough close to the surface: there may be foehn flow aloft and in the Wipp Valley (Mayr et al., 2004) while closer to the surface strong westerly winds are often observed in and around Innsbruck, so-called 'pre-foehn westerlies' (Seibert, 1985; Zängl, 2003), although these do not necessarily precede a breakthrough. For cases where the cold pool is partially eroded, intermittent foehn can occur in Innsbruck bringing periods of high wind speeds, intense turbulence and extreme spatial variability in flow and temperature. Thus, depending on the timing and location of the foehn breakthrough, there can be considerable differences in wind speed, wind direction, temperature and humidity within a few kilometres (Muschinski et al., 2021). Foehn in Innsbruck is most common in the afternoons (as the cold pool is often weakest or already dispersed at this time). Processes contributing to the onset and cessation of foehn in Innsbruck are explored in detail in Haid et al. (2020); Haid et al. (2021) and Umek et al. (2021).

Foehn has a marked impact on conditions in Innsbruck. Particularly during the colder months, foehn can cause the air temperature to increase by 5-15 °C and can thus play an important role in snowmelt and sublimation. Many foehn events stand out in timeseries data as high wind speeds accompanied by substantially enhanced air temperatures (Figure 3c, e). There is also a clear impact on turbulence: high air temperatures drive large negative $Q_H$, even in the city, and *TKE* values of 5-15 $m^2\ s^{-2}$ are not uncommon (Figure 3f-g). The large spread of *TKE* values (as well as wind speed and friction velocity) seen outside the summer months in Figure 5 is a result of foehn (either a complete breakthrough at the surface or 'pre-foehn' conditions strongly influenced by foehn).

### 5.4    Foehn case studies

Three case-studies involving foehn events are presented in Figure 7 and Figure 8. For all of these cases strong southerly winds at Steinach suggest almost continuous foehn in the Wipp Valley, supported by high foehn probabilities from the statistical model of Plavcan et al. (2014). In Innsbruck, there are periods when direct foehn





reaches the floor of the Inn Valley, when deflected foehn air reaches IAO and periods when foehn does not reach
the surface but nevertheless strongly influences conditions.
For 09 November to the evening of 12 November 2018 there is almost continuous foehn in the Wipp Valley with
some transient breaks or weakening indicated by small changes in wind direction and slight cooling (e.g. overnight
09-10 November 2018). On the afternoons of 10, 11 and 12 November, there is a clear example of a direct foehn
breakthrough at IAO, evident from the dramatic increase in temperature, the shift in wind direction from westerly
to southerly and the sudden increase in wind speed and *TKE* (Figure 7a-g). These events last a few (4-7) hours
each, during which the potential temperature at IAO matches the potential temperature in the Wipp Valley
(indicating that the atmosphere is well-mixed). Overnight the cold pool in the Inn Valley re-establishes and pre-
foehn westerly winds return near the surface, while foehn is still present aloft and in the Wipp Valley. On 09
November foehn does not reach IAO, possibly due to a more persistent cold pool resulting from the lower radiative
input on this day. However, the impact of foehn is still felt in Innsbruck with strong pre-foehn westerlies and large
*TKE*.
The situation is somewhat different for the April 2019 case study (Figure 7j-r). Foehn is continuously observed at
IAO from around midday on 22 April to midday on 26 April. The air temperature remains high throughout this
period (it is around 10 °C warmer overnight than without foehn on 21 and 27 April) and, in contrast to Figure 7b,
the potential temperature closely follows that in the Wipp Valley for the whole period. The wind speed is mostly
high and the wind direction is mainly southerly, but north-easterly winds are observed at times indicating that
southerly winds are not a necessary condition for foehn at IAO. Weak northerly or easterly winds are often a result
of foehn air being deflected from Nordkette before reaching IAO (Haid et al., 2020; Umek et al., 2021). Intense
turbulent mixing is evident from the high *TKE* values. Typically, $CO_2$ accumulates close to the surface overnight
leading to a peak in $CO_2$ mixing ratio ($r_{CO2}$) in the early morning hours before turbulent mixing increases and the
boundary layer begins to grow (Reid and Steyn, 1997; Schmutz et al., 2016). During foehn, however, the sustained
mixing prevents the usual build-up of $CO_2$ (and other pollutants) in the nocturnal boundary layer so that there is
very little diurnal variation and the usual early morning peak (visible on days without foehn overnight in Innsbruck)
is not seen (compare Figure 7g, p). Because of the increased mixing and influx of clean air during foehn, the
nocturnal $CO_2$ mixing ratios are generally lower during days with at least some foehn than on clear-sky days with
calm nights (e.g. compare with Figure 6).
For the third case study (Figure 8), data are presented for both IAO and FLUG to illustrate differences due to
location and surface cover. On 07, 08, 09, 10 and 13 April 2018, foehn reaches the Inn Valley during the daytime
and is interrupted during the night; on 11-12 April there is no apparent interruption of the foehn (note, again, the
flattened diurnal cycle of $r_{CO2}$). While the wind direction at IAO during the foehn periods on 07, 08 and 13 April
is consistently southerly, some easterly or northerly winds are observed on 09-12 April. At FLUG, south foehn
often appears as easterly winds since direct southerly flow is blocked in many cases by the mountains to the south
of the site. During the interruptions to foehn in the Inn Valley, the potential temperature at IAO and FLUG drops
below that in the Wipp Valley and pre-foehn westerlies are seen at both sites.
The increase in air temperature associated with the arrival of warm foehn air affects the near-surface temperature
gradient and thus the heat fluxes. Studies investigating the impact of downslope winds on snowmelt over prairies
in Alberta, Canada (MacDonald et al., 2018) and farmland in Hokkaido, Japan (Hayashi et al., 2005) reported
reduced or negative $Q_H$ and enhanced $Q_E$ during foehn-type winds. A similar effect is seen at FLUG. During foehn
the elevated air temperature can exceed the surface temperature and cause $Q_H$ to decrease and turn negative long
before sunset (Figure 8j, l). Enhanced $Q_E$ is sometimes seen accompanying the reduced $Q_H$ (e.g. 07 and 08 April,
Figure 8j). $Q_H$ at IAO is usually positive but during foehn or pre-foehn tends to be smaller and can even be negative.
Higher surface temperatures in the city mean that, even during foehn, $T_{air}$ rarely exceeds $T_{sfc}$ by much, so this effect
is somewhat reduced. However, during the colder months this smaller temperature gradient acts to reduce $Q_H$ and
can even result in negative $Q_H$ values, which are unusual for a city-centre site. Since evaporation is limited by the
availability of water (not energy), enhanced $Q_E$ is not observed during foehn at this urban site. However, at a
suburban site in Christchurch, New Zealand with greater moisture availability due to 56% vegetation cover,
negative $Q_H$ and enhanced $Q_E$ were observed during foehn (Spronken-Smith, 2002).



## 6    Radiation balance

### 6.1    Orographic shading

Similar to the way that buildings reduce the sky view factor in urban canyons (e.g. Johnson and Watson, 1984), the surrounding terrain reduces the sky view factor in valleys (Whiteman et al., 1989). This orographic shading effect is generally largest in deep, narrow valleys with a north-south orientation (e.g. Matzinger et al., 2003) but even in wider valleys local sunrise can be later and local sunset earlier compared to sunrise and sunset over flat terrain. At IAO sunrise occurs on average 45 minutes later and sunset 50 minutes earlier, although this varies throughout the year: the longest delay to sunrise is 90 minutes in winter, whereas the longest shift in sunset is 80 minutes in summer (Figure 9a-b). Since the orographic shading effect depends on the shape of the terrain relative to solar angle, for IAO the effect is smallest not in mid-summer when the sun is highest in the sky but when the solar azimuth angles around sunrise and sunset are aligned with the Inn Valley in spring and autumn. Although straightforward to calculate from a digital elevation map and solar angles (here determined using the R package solaR (Perpiñán, 2012)), orographic shading is highly spatially variable and for different locations within the city there can be differences in local sunrise/sunset times of some tens of minutes. Towards the edges of the city at the base of the north-facing slope, the total solar radiation receipt is reduced substantially.

The overall impact of orographic shading on the annual solar radiation receipt at IAO is small with around 2% less solar radiation received over the course of the year (this is an upper limit assuming clear-sky conditions and that incoming shortwave radiation, $K_\downarrow$, is zero before/after local sunrise/sunset; in reality there is a small diffuse component). The impact is larger in winter when daily solar radiation receipt is reduced by about 10%. Although the long-term impact of orographic shading is small at IAO, the sudden change in $K_\downarrow$ at local sunrise or sunset can be substantial (~100 W m$^{-2}$ within a few minutes, Figure 9c-d). A secondary shading effect is also observed: on mostly clear-sky days cumulus clouds often form above the mountain peaks during the afternoons, making sunset appear even earlier since the sun is blocked by clouds before it is blocked by terrain (Figure 9e-f). For a sloped grassland site in the Swiss Alps, a large and rapid drop in $K_\downarrow$ at local sunset has been shown to result in a drop in surface temperature of 10 °C in less than 10 minutes (Nadeau et al., 2013). Such an effect would likely be smaller in urban areas due to the large thermal capacity of building materials and the additional anthropogenic heat supply (which does not stop abruptly at sunset). Note that for sloping sites $K_\downarrow$ may far exceed that over flat terrain (depending on the slope and aspect of the surface relative to the direct beam solar radiation), the shape of the diurnal cycle can be very asymmetrical and its peak may be shifted earlier or later (Matzinger et al., 2003; Nadeau et al., 2013).

The surrounding terrain also affects longwave exchange. The smaller the sky view factor and greater the proportion of the radiometer field of view taken up by relatively warm valley slopes, as opposed to cold sky, the larger the observed incoming longwave radiation $L_\downarrow$ (Whiteman et al., 1989). It was not possible to quantify this effect here, however it is expected that $L_\downarrow$ is larger in Innsbruck than over an equivalent site in flat terrain. The principle is analogous to enhanced $L_\downarrow$ in street canyons due to emissions from surrounding buildings (Oke, 1981).

### 6.2    Atmospheric transmissivity

Atmospheric transmissivity was investigated using a clearness index which expresses the ratio of observed $K_\downarrow$ to incoming shortwave radiation at the top of the atmosphere $K_{\downarrow TOA}$ (calculated assuming a solar constant of 1367 W m$^{-2}$ (Peixoto and Oort, 1992)). The mean value of the clearness index for the study period is 0.45 ($K_\downarrow > 5$ W m$^{-2}$). Considering clear-sky days only (see Appendix A), the midday (11:00-15:00 CET) clearness index is 0.74 and shows some seasonal variability (Figure 10a), being highest in early spring at 0.78 when the boundary layer height is considerable and water vapour content is low, and lower from summer through to winter at 0.68-0.73. This is attributed to increased moisture and aerosols in the atmosphere in summer, and reduced boundary layer growth and increased pollutant concentrations in winter. Although air quality is a major issue for Alpine valleys, it is even more critical for larger cities in complex terrain, such as Mexico City (Velasco et al., 2007) where air pollution was found to deplete $K_\downarrow$ by 22% compared to a rural reference site (Jáuregui and Luyando, 1999).

### 6.3    Albedo

At IAO the average midday (11:00-15:00 CET) albedo is 0.16. This is within the range of values previously obtained for European cities, such as 0.08 in Łódź (Offerle et al., 2006), 0.11 in Basel (Christen and Vogt, 2004), 0.11 and 0.14 for two sites in London (Kotthaus and Grimmond, 2014b) and 0.16-0.18 in Marseille (Grimmond et





al., 2004). For one of the London sites and the Marseille site, light-coloured roof surfaces below the radiometers
led to observed albedos that were probably higher than the bulk local-scale albedo. Similarly, at IAO the radiometer
source area is mainly comprised of light-coloured roof and concrete surfaces, and thus the measured outgoing
shortwave radiation ($K_\uparrow$) is probably slightly higher than the local-scale average.
As there is negligible vegetation within the radiometer field-of-view, $\alpha$ is fairly constant all year-round except
when snow covers the surface (Figure 3a). In the city, snow is usually cleared from roads within a few hours and
melts quickly on roofs. Thus, in contrast to the rural surroundings, $\alpha$ at IAO usually decreases quickly and returns
to the lower non-snow-covered values after a few days (although heavy snow cover in January 2019 remained on
rooftops and uncleared areas for longer). Snow cover at IAO and FLUG was determined from visual inspection of
webcam images from the university and Innsbruck Airport. When there is a thick covering of fresh snow, $\alpha$
increases to about 0.4 at IAO and 0.7 at FLUG. This difference between urban and rural values is partly due to the
large proportion of non-snow-covered vertical walls seen by the IAO radiometer and partly due to the short
duration of pristine snow cover in the city (before clearing or melting occurs). The IAO values obtained here are
also lower than for a suburban site in Montreal (0.6-0.8) where the duration of snow cover is much longer (Järvi
et al., 2014). During non-snow periods, $\alpha$ at FLUG is about 0.21 (September-May dataset), similar to that observed
at the Neustift FLUXNET site – a grassland site in the nearby Stubai Valley (Hammerle et al., 2008).
Although there is little seasonal trend in $\alpha$ at IAO, there is considerable variability associated with solar elevation
angle ($\theta_{elev}$), azimuth angle, cloud cover and rainfall. Due to the location of the IAO radiometer at the south-eastern
edge of the building, shading of the street below during the afternoon causes asymmetry in the diurnal cycle of $\alpha$.
For similar $K_\downarrow$ and $\theta_{elev}$, $\alpha$ is up to about 0.05 smaller in the afternoon than the morning. This difference is greatest
during sunny conditions and almost disappears for high elevation angles (Figure 10b). $\alpha$ is much larger under
clear-sky conditions (0.15-0.20) compared to cloudy conditions (close to 0.10). Shortly following rainfall the
albedo drops by about 0.03 and steadily rises over the following 6 hours as the surface dries (Figure 11a).

## 6.4     Radiative fluxes


Figure 12 shows the seasonal and diurnal patterns and interannual variations in radiation and energy fluxes at IAO
for the four-year study period. For comparison, data for FLUG are also shown for the period that the site was
operational. Although interannual variability in the radiation and energy fluxes is generally small (Figure 12l-u),
some differences can be identified. The shortwave radiation components show the most variability and the
particularly cloudy months of September 2017 and May 2019 can be clearly seen in Figure 12l. The effect of this
reduced $K_\downarrow$ is also visible in $K_\uparrow$, the net radiation ($Q^*$) and the sensible heat flux (Figure 12m, p, r) and to a lesser
extent in the outgoing longwave radiation ($L_\uparrow$) (Figure 12o). Except for cloudy September 2017 and February
2018, the fluxes at IAO for the period that FLUG was operational are very similar to those over the whole dataset.
Of the four radiation components, $K_\downarrow$ has the largest annual cycle, providing an average of 38 W m$^{-2}$ (3.3 MJ m$^{-2}$
day$^{-1}$) in December and 289 W m$^{-2}$ (25.0 MJ m$^{-2}$ day$^{-1}$) in June at IAO. This is the main energy input to the surface.
As the albedo is fairly small and constant throughout the year, the net shortwave radiation ($K_\downarrow - K_\uparrow$) and the net
all-wave radiation $Q^*$ during daytime closely follow $K_\downarrow$. Comparing radiative fluxes for the same time period, $K_\downarrow$
at IAO and FLUG is very similar (for 30-min values the square of the correlation coefficient $r^2 = 0.97$), as expected
given the close geographical proximity of the sites, but the extended period of snow cover at FLUG leads to much
higher $K_\uparrow$ than at IAO during winter (Figure 12b). This also leads to a substantial difference in $Q^*$ (Figure 12e).
Average $Q^*$ for February 2018 was 27 W m$^{-2}$ (2.3 MJ m$^{-2}$ day$^{-1}$) at IAO compared to only 2 W m$^{-2}$ (0.2 MJ
m$^{-2}$ day$^{-1}$) at FLUG.
Outgoing longwave radiation follows a clear diurnal cycle all year round, which is of considerable amplitude (>
100 W m$^{-2}$) during summer. Only on days with very little solar radiation does $L_\uparrow$ depart from this pattern. $L_\uparrow$ is
higher in summer than in winter because the surface temperature is higher. Compared to the grassland site, $L_\uparrow$ is
larger in the city and remains higher in the evenings and overnight (Figure 12d), as stored heat is slowly released
from the urban fabric and anthropogenic activities continue to provide energy. Incoming longwave radiation has
a less repeatable diurnal pattern, as it is influenced to a much greater extent by cloud cover and thus is more
variable, particularly during winter. Only on clear-sky days does $L_\downarrow$ follow a smooth diurnal cycle. While the net
shortwave radiation ($K_\downarrow - K_\uparrow$) is positive all year round (zero at night), the net longwave radiation ($L_\downarrow - L_\uparrow$) is




negative all day and all year as $L_\uparrow$ almost always exceeds $L_\downarrow$ (except for a few cases (<2%) which occur mainly
when the surface is covered with snow).
From November to January, the net longwave loss is similar to the net shortwave gain and daily $Q*$ is close to
zero (Figure 12p). The magnitudes of the net longwave loss and the net shortwave gain both increase towards
summer, but the net shortwave gain increases faster resulting in a substantial net radiative energy input. Average
$Q*$ is -4 W m$^{-2}$ (-0.4 MJ m$^{-2}$ day$^{-1}$) in December and 159 W m$^{-2}$ (13.7 MJ m$^{-2}$ day$^{-1}$) in June with typical peak
daytime values of 120 W m$^{-2}$ in December and 540 W m$^{-2}$ in June/July. At night $Q*$ is negative as a result of
longwave cooling, more so in summer than winter, and more so at the urban site than the rural site (Figure 12e).

## 514    7    Energy balance

### 515    7.1    Anthropogenic heat flux

In the urban environment, the available energy is supplemented by additional heat released from human activities
(Oke et al., 2017). This anthropogenic heat flux ($Q_F$) includes energy use in buildings ($Q_B$) and for transportation
($Q_V$) as well as energy from human metabolism ($Q_M$). Here, $Q_F$ was estimated as described in Appendix B using a
typical inventory approach similar to that at other sites (e.g. Sailor and Lu, 2004). For this area of Innsbruck, $Q_F$
is estimated to provide an average of 9-19 W m$^{-2}$ per day (of which approximately 1 W m$^{-2}$, 5 W m$^{-2}$ and 3-14 W
m$^{-2}$ are from $Q_M$, $Q_V$ and $Q_B$, respectively). $Q_F$ is highest in the coldest months when the demand for building
heating is greatest and most of the interannual variability (Figure 12q) is due to temperature. However, $Q_F$ is lower
than would otherwise be expected in March-April 2020 and November 2020-January 2021 due to a substantial
reduction in traffic during Coronavirus lockdowns (no information was available about how building energy use
changed over this period, however). The magnitude of $Q_F$ for this site in the centre of a small city lies between the
typical values of 5-10 W m$^{-2}$ found for suburban sites (e.g. Pigeon et al., 2007; Bergeron and Strachan, 2010; Ward
et al., 2013) and the much higher values (> 40 W m$^{-2}$) obtained for central sites in larger and more densely built
cities (e.g. Ichinose et al., 1999; Nemitz et al., 2002; Hamilton et al., 2009). $Q_F$ is about 20% less on non-working
days (i.e. weekends and holidays) compared to working days.
Although the anthropogenic energy input is small compared to the net radiation in summer, $Q_F$ becomes a more
significant source of energy in winter. During winter daytime, $Q_F$ accounts for around 20% of the available energy
(i.e. $Q* + Q_F$). On a 24-h basis, $Q_F$ increases the available energy from close to zero to around 20 W m$^{-2}$ in winter
(Figure 12p, q), which helps to maintain a positive sensible heat flux all year round (Figure 12g, r). Furthermore,
$Q_F$ affects the difference in available energy between the city and rural surroundings (where $Q_F$ is assumed to be
zero), enhancing spatial variability in turbulent fluxes which could impact local circulation patterns and cold pool
evolution.

### 537    7.2    Net storage heat flux

The net storage heat flux, $\Delta Q_S$, is another important term in the urban energy balance but very difficult to measure
directly (Offerle et al., 2005a). Two commonly used approaches are used to estimate $\Delta Q_S$ here. The first approach
is the Objective Hysteresis Model (OHM) of Grimmond et al. (1991) which calculates $\Delta Q_S$ from $Q*$, the rate of
change of $Q*$ and empirical coefficients for different land cover types (see Appendix C for details). The second
approach estimates $\Delta Q_S$ as the residual (RES) of the energy balance ($\Delta Q_S = Q* + Q_F - Q_H - Q_E$). Both approaches
have limitations. OHM relies on coefficients derived from a handful of observations or simulation studies and does
not account for changes in surface conditions (e.g. soil moisture), while the residual approach ignores advection,
assumes the energy balance is closed (which it likely is not) and errors in the other energy balance terms collect
in the estimate of $\Delta Q_S$. Nevertheless, for IAO the two storage heat flux estimates are in remarkably good agreement
(Figure 12i). The magnitude of $\Delta Q_{S\_OHM}$ is slightly smaller than $\Delta Q_{S\_RES}$ but the seasonal and diurnal cycles are
well represented overall. This contrasts with two UK sites where OHM was found to perform poorly in winter,
underestimating the daytime storage release in central London and overestimating the storage release over the
whole day in suburban Swindon (Ward et al., 2016).
During the day, when heat is being stored in the large thermal mass of the buildings and roads, $\Delta Q_S$ is large and
positive (with peak daytime values of 230 W m$^{-2}$ in summer and 40 W m$^{-2}$ in winter). At night, stored heat is
released to the atmosphere. The substantial negative $\Delta Q_S$ ($\approx$ -50 W m$^{-2}$) during night-time supports the positive
sensible heat fluxes and appreciable longwave cooling. The larger the available thermal mass (i.e. the more densely





built the city), the greater the potential to store heat (hence, the ground heat flux at FLUG is much smaller in
comparison). From April to August, more heat is stored in the urban fabric than released, whereas the opposite is
true from October to February (Figure 12t). In theory, the losses and gains should cancel over the year but here
$\Delta Q_{S\_OHM}$ and $\Delta Q_{S\_RES}$ yield a small net gain (2.4 and 6.5 W m$^{-2}$), similar to previous studies (Grimmond et al.,
559 1991).

### 7.3 Turbulent heat fluxes

As has been observed at other densely built city-centre sites, such as Basel (Christen and Vogt, 2004), Łódź
(Offerle et al., 2005a) and London (Kotthaus and Grimmond, 2014a), the sensible heat flux generally remains
positive all day and all year round at IAO (Figure 12g, r). Daily average $Q_H$ ranges from 28 W m$^{-2}$
(2.4 MJ m$^{-2}$ day$^{-1}$) in December to 77 W m$^{-2}$ (6.7 MJ m$^{-2}$ day$^{-1}$) in June. Peak average daytime $Q_H$ is highest in
April and remains fairly constant above 180 W m$^{-2}$ April to August. Average night-time values are positive at
around 10-15 W m$^{-2}$. The latent heat flux is much smaller (Figure 12h, s) with daily average values from 7 W m$^{-2}$
(0.6 MJ m$^{-2}$ day$^{-1}$) in December-January to 26 W m$^{-2}$ (2.3 MJ m$^{-2}$ day$^{-1}$) in June-July. Peak daytime values average
around 55 W m$^{-2}$ in summer (i.e. less than a third of peak $Q_H$ values) and $Q_E$ remains small and positive overnight
(4-8 W m$^{-2}$). Thus, most of the available energy is directed into either heating the atmosphere ($Q_H$) or heating the
surface ($\Delta Q_S$).

### 7.4 Energy partitioning

To facilitate comparison between sites, energy fluxes are often considered relative to the net radiation. The values
of the resulting ratios depend on the time of day considered (e.g. midday, daytime, 24-h) as well as season (Figure
13). At IAO month-to-month variation in the energy flux ratios is quite small from late spring to early autumn but
the ratios change more quickly (and are more variable) in winter when the energy supplied is smaller and days are
shorter. $Q_H/Q^*$ is 0.42, $\Delta Q_S/Q^*$ is 0.40 and $Q_E/Q^*$ is 0.14 on average during summer daytime. During winter
$Q_H/Q^*$ is larger (around 0.65) and $\Delta Q_S/Q^*$ is smaller, partly due to the additional anthropogenic energy input from
building heating and the tendency for heat that has been stored in the urban environment to be released. Daytime
$Q_E/Q^*$ is lowest in February-April at < 0.1 and highest in August-October at 0.14-0.15, roughly corresponding to
seasonal variability in rainfall (Figure 3h); the small amount of vegetation likely makes only a minor contribution
to increased evapotranspiration during summer. The Bowen ratio ($\beta = Q_H/Q_E$) is largest between January and April
at 5.4-5.5 and decreases in the summer months to a minimum of 2.5 in August (average daytime values).
As there are few vegetated or pervious surfaces in the centre of Innsbruck, there is little possibility for rainwater
to infiltrate and be stored in the urban surface so the effect of rainfall on the surface energy balance is short-lived.
When surfaces are wet shortly after rainfall, enhanced evaporation rates are observed (Figure 11b-c). Directly
following rainfall $Q_E/Q^*$ is around 0.35 and $\beta$ around 1.4; over the next 6-12 hours $Q_E/Q^*$ falls to around 0.13 and
$\beta$ increases above 3.5. These values represent averages over the whole dataset; naturally there is variation in
magnitude and drying time according to season, weather conditions and the amount of precipitation. However, the
impact of rainfall seems to be quite short in Innsbruck. At (sub-)urban sites with a greater proportion of pervious
surfaces the process can take several days (Ward et al., 2013), while in central London a slightly longer drying
time of 12-18 h was reported (Kotthaus and Grimmond, 2014a). The shorter time suggested for Innsbruck may be
due to the abundance of summer precipitation which would be expected to evaporate quickly from hot surfaces.
Despite similar $Q^*$ at IAO and FLUG (except during snow cover), the different surface characteristics lead to very
different energy partitioning. At FLUG, evapotranspiration from the grass means large $Q_E$ values are measured in
spring and autumn. Because more energy is used for $Q_E$, $Q_H$ is much smaller (Figure 12g-h). Daytime $Q_H/Q^*$ and
$Q_G/Q^*$ are around 0.1-0.2 in spring and autumn, while $Q_E/Q^*$ exceeds 0.5 in April-May. As expected for a non-
urban site, $Q_H$ is negative during the night (Figure 12g).
It has been possible to link the average energy partitioning observed in previous urban studies to surface
characteristics (typically land cover) through simple empirical relations (e.g. Grimmond and Oke, 2002; Christen
and Vogt, 2004), although these are mainly based on summertime data when most field campaigns took place.
Almost all urban studies conclude that although $Q_E$ can be small it is not negligible and during daytime $Q_E/Q^*$ is
typically between 0.1 and 0.4. The lower end of this range represents urban sites, such as IAO, with little vegetation
(e.g. sites U1 and U2 in Basel (Christen and Vogt, 2004), Marseille (Grimmond et al., 2004) or Shanghai (Ao et
al., 2016)), while the upper end corresponds to more vegetated areas often with irrigation (e.g. suburbs of North





American cities (Grimmond and Oke, 1995; Newton et al., 2007)). The values obtained at IAO for $Q_H/Q^*$ and
$\Delta Q_S/Q^*$ are also within the range expected from previous studies: (0.2-0.5 during summer, with $Q_H/Q^*$ towards
the lower end of this range for vegetated and irrigated sites). At IAO slightly more energy is directed into $Q_H$ than
$\Delta Q_S$, as was also found for urban sites in Basel (Christen and Vogt, 2004). In Marseille, $Q_H/Q^*$ was much larger
than $\Delta Q_S/Q^*$ at 0.69 and 0.27, respectively (Grimmond et al., 2004), while in Mexico City the opposite was found
(Oke et al., 1999). At IAO, observed $\beta$ is relatively high (daytime median $\beta$ is 3.7) compared to previous studies
but only slightly higher than would be expected (and well within the scatter) given the vegetation fraction (Figure
14a). Hence, it can be concluded that, on average, energy partitioning at this site in highly complex terrain does
not deviate substantially from the existing urban literature.
At shorter timescales, however, the energy balance terms are impacted by Innsbruck's orographic setting. An
interesting feature of observed $Q_H$ at FLUG is the unusual shape of the diurnal cycle, particularly in April and May
(Figure 12g). Rather than peaking close to noon, $Q_H$ peaks in the morning and then becomes appreciably negative
in the afternoon, while $Q_E$ remains large and positive until sunset. Close inspection of the time-series reveals that
this is largely a result of warm foehn air (i.e. warmer than the surface beneath) reaching FLUG in the afternoon
(Figure 8j, Section 5.4) and, since foehn occurred frequently during Spring 2018, this pattern shows up in the
monthly averages. However, similar behaviour is also seen on valley-wind days (Figure 6j) and has been observed
at other rural sites in the Inn Valley as well (Vergeiner and Dreiseitl, 1987; Babić et al., 2021; Lehner et al., 2021).
At IAO, the much larger $Q_H$ and smaller $Q_E$ means that a similar change in sign of $Q_H$ during the afternoon is not
seen, but $Q_H$ does rise earlier in the day and peak first (just before or around solar noon) while $Q_E$ reaches its
maximum later in the day (after solar noon) and remains at moderate values until the evening. As a result, the
diurnal cycle of the Bowen ratio at IAO is asymmetrical, seen most clearly in summer (Figure 12j).
The reason for this phase shift between $Q_H$ and $Q_E$ is not fully understood. Frequent afternoon thunderstorms seem
to enhance $Q_E$ during summer afternoons, but the trend remains if times with rain and shortly following rain are
excluded. A larger vapour pressure deficit in the afternoons acting to increase $Q_E$ could also be a contributing
factor. Air temperature exceeding surface temperature during foehn (at both IAO and FLUG) or during the
afternoon on some valley-wind days (at FLUG) also plays a role. The behaviour does not appear to be related to
differences in radiative input (since the shift between $Q_H$ and $Q_E$ is also seen in the ratio of the turbulent fluxes to
$Q^*$), nor changing source area characteristics.
Due to the temporal patterns in wind direction at this complex-terrain site (Section 5.1), the source area itself varies
systematically with time of day and season, being located west of IAO (with a slightly larger vegetation fraction,
a larger proportion of water and a smaller building fraction) for down-valley winds during night-times and winter
months, and east of IAO (towards the city centre) for up-valley winds during summer daytimes. There is no clear
evidence of spatial variations in energy partitioning as a result of surface cover variability around the site, although
some variation with wind direction is observed as a result of differences in weather conditions (e.g. fair weather
tends to coincide with daytime up-valley winds). The slightly higher vegetation fraction for down-valley winds is
not large enough to generate a discernible increase in evaporative fluxes for this wind sector. Furthermore, the
river, despite its proximity, does not provide a strong evaporative flux that is detected by the instruments. Similar
results were found in central London, where a major river passes close to the measurement site yet appears not to
contribute to the observed moisture flux (Kotthaus and Grimmond, 2014b). Perhaps an internal boundary layer
forms over the river which does not reach the instrument height, or the low water temperature could limit
evaporation (Sugawara and Narita, 2012).
**8    Carbon dioxide exchange**
The observed $CO_2$ fluxes ($F_{CO2}$) at this city-centre site are dominated by anthropogenic emissions. $F_{CO2}$ is positive
throughout the day and all year round (Figure 12k, v). Similar to other city-centre sites (e.g. Nemitz et al., 2002;
Björkegren and Grimmond, 2017; Järvi et al., 2019), the highest values are generally observed during the middle
of the day, but the shape of the diurnal cycle at IAO varies throughout the year. In the winter months the flux is
slightly higher either side of midday and more closely resembles the typical double-peaked diurnal cycle often
attributed to rush-hour activities, while in summer the peak appears to be shifted towards the afternoon.





In contrast, there is clear photosynthetic uptake at the grassland site (FLUG) during the growing season (Figure
12k). Here, daytime $F_{CO2}$ during spring and autumn follows a typical light-response curve when plotted against
photosynthetically active radiation, $PAR$ (Figure 15a), estimated as a proportion (0.47) of $K_\downarrow$ (Papaioannou et al.,
1993). At IAO there is little dependence of $F_{CO2}$ on $PAR$ (for small $PAR$ the tendency for $F_{CO2}$ to increase as $PAR$
increases is because both anthropogenic activity and $K_\downarrow$ are largest in the middle of the day). At FLUG, the increase
in night-time $F_{CO2}$ with temperature (Figure 15b) suggests soil respiration is responsible for increased emissions
during night-time in spring and autumn (Figure 12k). At IAO, any contribution of soil respiration is minor. Indeed,
the opposite behaviour is seen, and $CO_2$ emissions decrease with increasing temperature as demand for building
heating falls. Although anthropogenic emissions far outweigh any biogenic contributions to the observed $CO_2$
fluxes, it is possible to identify biogenic signals in other gases at IAO (Karl et al., 2018; Kaser et al., 2021).
To further explore the anthropogenic controls on $F_{CO2}$ at IAO, the dependence on air temperature is shown at daily
and monthly timescales in Figure 15c-d. At the daily timescale, there is a clear linear decrease in $F_{CO2}$ with
increasing temperature up to around 18 °C, above which $F_{CO2}$ remains constant. This type of behaviour suggests a
substantial contribution of fuel combustion for building heating to observed $F_{CO2}$ (Sailor and Vasireddy, 2006;
Bergeron and Strachan, 2011). On average the data suggest that fuel combustion for space heating releases an extra
0.5 µmol m$^{-2}$ s$^{-1}$ $CO_2$ for every 1 °C decrease in temperature, although this is thought to be an underestimate as the
seasonal variability in amplitude is smaller than might be expected (see below). Assuming negligible contribution
from photosynthesis or soil respiration, the temperature-independent anthropogenic emissions amount to an
average of 11.4 and 8.0 µmol m$^{-2}$ s$^{-1}$ on working and non-working days, respectively. These approximately
temperature-independent emissions are attributed to human metabolism, fuel combustion for transport and fuel
combustion in buildings that is not associated with space heating (e.g. for cooking or heating water).
The scatter seen in Figure 15c arises from various factors, including the changing measurement source area,
variability in human behaviour and the impact of weather conditions besides temperature (such as snow, rain or
solar radiation affecting people's perception of temperature). The timing of unusually cool or warm spells affects
energy consumption (e.g. a cold spell in September is likely associated with lower emissions than for the same
temperature in December because people may not have switched on their heating yet). Similarly, high temperatures
during foehn also contribute to deviations, particularly during winter (days with foehn tend to have higher
emissions than would be expected given the temperature). Observed $F_{CO2}$ is higher on working days compared to
non-working days by about 4 µmol m$^{-2}$ s$^{-1}$. Although working and non-working days have already been separated,
emissions on Sundays tend to be lower than on Saturdays and there is also some variability Monday-Friday. During
the Coronavirus restrictions, reduced emissions resulted in generally lower observed $F_{CO2}$, particularly on working
days (see also Lamprecht et al., 2021).
The temperature dependence of building heating demand explains most of the monthly variation in observed $F_{CO2}$
(Figure 15d, square of the correlation coefficient $r^2 = 0.57$). Reduced traffic during the periods with the strictest
Coronavirus restrictions in March-April 2020 and November 2020-February 2021 means observed $F_{CO2}$ is also
towards the bottom of the distribution for these months (see also Figure 12v). The highest monthly $F_{CO2}$ was
recorded for February 2018 and is considerably higher than expected given the average monthly temperature. This
is attributed to a period of very cold weather towards the end of the month (Figure 3c) which coincided with
easterly winds.
A marked difference in observed $CO_2$ fluxes with wind direction is seen at IAO. Fluxes from the eastern sector
(60-120°) are about twice as high as those from the western sector (210-270°). Monthly diurnal cycles are
considered to avoid biases by season and by time of day (Figure 16a, b), although easterly winds are still associated
with more unstable conditions. Although the land cover composition is quite similar for these two main sectors
(Section 3), the eastern sector is more densely built with a larger proportion of buildings and roads and *busier*
roads (including a crossroads close to the site), whereas the western sector contains fewer roads, more widely
spaced institutional buildings and more vegetation and water (Figure 2). The observed $F_{CO2}$ data represent a
combination of the spatiotemporally varying contributions of anthropogenic emissions, the variation in the flux
footprint with season and with time of day and the level of turbulent mixing. The strong seasonal and diurnal
dependence of wind direction (Section 5.1) must be considered when interpreting the dataset, as characteristic
features arise from a mixture of temporal changes in sources and sinks combined with differences in spatial
sampling due to the changing source area. For example, the shift in peak $CO_2$ fluxes to the afternoon that is





particularly evident in summer is likely due to the diurnal wind shifting from westerly (low emissions) to easterly
(high emissions) since this asymmetry is not seen in the diurnal cycles of west and east sectors separately (Figure
16a). Nor is this behaviour seen in emissions modelled using a statistical inventory approach (analogous to the
estimation of $Q_F$, Appendix B) which accounts for temporal variability in human activities but not spatial
variability around the tower. Moreover, in winter, when westerly winds are more common, the observed data are
more representative of the lower emissions from the western sector, whereas in summer the afternoon data are
more representative of higher emissions from the eastern sector. This bias in source area sampling probably results
in lower wintertime fluxes and higher summertime averages compared to if the observations were evenly
representative of the source area. This leads to slightly smaller seasonal variation (and hence a weaker temperature
dependence in Figure 15c-d), as the higher emissions in winter are partly compensated by a greater frequency of
westerly winds leading to lower observed fluxes (and the opposite situation in summer). Additionally, the easterly
winds during the period of cold weather in February 2018 further enhanced observed $F_{CO2}$ during this period
compared to the source-area average.
Using median diurnal cycles to gap-fill the observations (Järvi et al., 2012) gives an annual total $CO_2$ flux of
5.1 kg C m$^{-2}$ y$^{-1}$ (varying between 4.3 and 6.0 kg C m$^{-2}$ y$^{-1}$ for the four May-to-May twelve-month periods in the
study period). For comparison, annual uptake at the nearby Neustift grassland site is 0.018 kg C m$^{-2}$ y$^{-1}$ (Wohlfahrt
et al., 2008). The annual $CO_2$ flux at IAO is well within expectations given the proportion of vegetation (Figure
14b). Similar annual totals (4.9-5.6 kg C m$^{-2}$ y$^{-1}$) and vegetation fractions (12-29%) were found for Basel (Schmutz
et al., 2016), Beijing (Liu et al., 2012), Helsinki (Järvi et al., 2019), Heraklion (Stagakis et al., 2019) and Montreal
(Bergeron and Strachan, 2011). In Montreal, the annual total is higher than suggested by the vegetation fraction
alone, possibly due to the cold climate (as is also the case for Vancouver (Christen et al., 2011)), while in
Heraklion, where space heating emissions are small, the annual total is lower. Considering the eastern and western
sectors separately at IAO gives annual totals of 7.0 and 3.3 kg C m$^{-2}$ y$^{-1}$, respectively. These values are also within
the range suggested by previous studies if the proportion of vegetation and water in the source area is considered
(10/28% for the eastern/western sectors). Even a small amount of vegetation or open water can make a substantial
difference to the emissions in city centres, as it is not only photosynthetic uptake by vegetation, but also the absence
of roads or buildings (which would contribute substantially to the emissions) associated with vegetated and water
surfaces, that is relevant.
Modelled $CO_2$ emissions (Figure 16b, d) result in a similar annual total of 5.0 kg C m$^{-2}$ y$^{-1}$ and suggest that human
metabolism accounts for 13% of the annual total emissions, traffic 35% and building energy use 53%. However,
these contributions vary considerably with time of year. On a daily basis, human metabolism contributes around
1.7 µmol m$^{-2}$ s$^{-1}$, fuel combustion for transport around 4.5 µmol m$^{-2}$ s$^{-1}$ and building energy use around
1.6 µmol m$^{-2}$ s$^{-1}$ in summer and 13 µmol m$^{-2}$ s$^{-1}$ in winter. In sum, daily total emissions are around 8 µmol m$^{-2}$ s$^{-1}$
in summer and 19 µmol m$^{-2}$ s$^{-1}$ in winter (Figure 16d). The modelled emissions suggest working/non-working day
differences are mainly due to traffic but building energy use also contributes during winter. The reasonable
agreement between modelled $CO_2$ emissions and observed $F_{CO2}$ give confidence that the analogously calculated
anthropogenic heat flux is an appropriate estimate for the study area. The model seems to overestimate emissions
from building heating in winter (Figure 16d) but this may partly result from the prevalence of westerly winds and
associated underestimation of observed $F_{CO2}$ compared to the source-area average. Future work will address the
fine-scale spatial and temporal variability in emissions around the tower in more detail.
**9    Impact of flow regime on near-surface conditions**
Having examined the climatology at IAO and explored the controls on the energy and carbon exchange, this section
summarises the effects of complex terrain flows on near-surface conditions through comparison of valley-wind
days, foehn events and pre-foehn conditions. As has been shown above, although a twice-daily wind reversal is
frequently observed in and around Innsbruck (Section 5.1), there are very few examples of purely thermally driven
valley-wind days. Similarly, many different types of foehn can occur with different characteristics and there is
often interaction with other types of flow. Given these complexities, a manual classification of different flow
regimes was judged to be the most useful approach for the purposes of this analysis (see Appendix A for details).
While several case studies are presented above, Figure 17 summarises the impact of the valley-wind circulation,
foehn events and pre-foehn conditions on near-surface conditions at IAO. Note that because the flow regimes



occur under different synoptic conditions, the first two columns group data from different times of day (and times
of year), while the third column helps to minimise the impact of diurnal and seasonal trends (although some lines
are incomplete as the different regimes do not necessarily occur at all times in all seasons, for example during
winter nights pre-foehn is more common than foehn).
During foehn $TKE$ and gust speeds are substantially higher than during non-foehn highlighting that foehn is
associated with intense turbulence as well as strong winds. $TKE$ during foehn is typically between 4 and 9 m$^2$ s$^{-2}$,
compared to days with a valley-wind reversal when $TKE$ reaches a maximum of 2-3 m$^2$ s$^{-2}$ during the up-valley
flow (Figure 17u). For pre-foehn conditions, wind speeds are comparable to those during foehn but $TKE$ and gust
speeds are lower (though still much higher than on valley-wind days). Up-valley wind speeds often reach 2-4 m s$^{-1}$
in spring and summer, slightly lower than median wind speeds during foehn, and down-valley winds are weak.
As valley-wind days are driven by the heating of the valley atmosphere and often occur on fair-weather days, the
amplitude of the diurnal cycles of $T_{air}$ and $T_{sfc}$ as well as the difference $T_{sfc} - T_{air}$ is large and, particularly during
the middle of the day, the surface is substantially warmer than the air (Figure 17y). Valley-wind days tend to have
large positive $Q_H$ which peaks early in the day (also evident in the ratio $Q_H/Q^*$, Figure 17z, A). During foehn, the
air temperature is substantially higher, especially during autumn and winter (Figure 17n), and the diurnal cycle is
much weaker (Figure 17x) since nocturnal cooling is considerably reduced if foehn continues through the night
(Figure 7k; Figure 8b, k). Enhanced $T_{air}$ during foehn frequently (54% of the time) exceeds the surface temperature
and results in negative sensible heat fluxes (mainly in autumn and winter but also during the night in spring, Figure
17p, z). In these cases, the influence of foehn overcomes the influence of the urban surface (which usually
maintains positive $Q_H$). Only 9% of $Q_H$ data are negative at IAO, and 44% of these occur during times classified
as foehn or pre-foehn. Interestingly, pre-foehn conditions tend to have similarly low $Q_H$ as for foehn conditions
(compare similar times of day and times of year in Figure 17z) despite lower air temperatures and a smaller
temperature difference between surface and near-surface atmosphere. This may be a result of warm foehn air being
brought down to the surface as the Inn Valley cold pool is eroded. Despite the high variability during foehn and
pre-foehn, the effect of reduced $Q_H$ during daytime and negative $Q_H$ during night-time can be seen in the ratio
$Q_H/Q^*$ (Figure 17A). For foehn conditions daytime $Q_H/Q^*$ is slightly smaller in spring and autumn but much
smaller in winter (0.2-0.4), and night-time $Q_H/Q^*$ changes from around -0.5 to above 0.5 (as both $Q_H$ and $Q^*$ are
negative). There is some evidence for slightly enhanced latent heat fluxes at IAO during foehn compared to valley
wind days (Figure 17B, C) but this effect is minor (smaller than at FLUG), probably because of the lack of available
water at IAO.
The intense mixing that accompanies foehn consistently maintains a low $CO_2$ mixing ratio. The low values of $r_{CO2}$
observed during foehn are similar to those during daytime on valley-wind days in spring and summer with
considerable thermal mixing and well-developed boundary layers (Figure 17D). The dynamical mixing of foehn
maintains low $r_{CO2}$ also during the night, in contrast to nights following valley-wind days when strong cooling and
low wind speeds lead to high $r_{CO2}$. Thus, foehn can be an important means of exchanging the air mass in the valley,
particularly for urbanised valleys in autumn and winter when the trapping and build-up of emissions can be
problematic. On the other hand, long-range transport and subsidence during foehn can increase levels of other
atmospheric constituents such as ozone (Seibert et al., 2000). There was no discernible impact of flow conditions
on the observed net $CO_2$ exchange, in accordance with Hiller et al. (2008) who measured $CO_2$ fluxes from a
grassland site in an Alpine valley and also concluded there was no obvious differences in the $CO_2$ uptake observed
during different wind regimes (foehn, valley-wind and persistent up-valley wind).
**10    Summary and conclusions**
Cities in mountainous terrain are subject to extreme and challenging conditions. For Innsbruck, heat stress in
summer, heavy snowfall, icing and avalanches in winter, flooding, downslope windstorms and air quality are all
relevant issues. Understanding the underlying physical processes and their interactions is key to better predicting
the occurrence, location and magnitude of such conditions. Moreover, knowledge of how cities in complex terrain
are similar to and different from cities in flat terrain is crucial to avoid inadvertent harmful effects that can result
from attempts to mitigate climate issues. For such process studies (as well as for evaluation of numerical models),
direct measurement techniques such as eddy covariance are extremely valuable.





Four years of energy and carbon dioxide fluxes from the Alpine city of Innsbruck are presented and analysed. This
study constitutes the first multi-year climatology of turbulence measurements from an urban area in highly
complex terrain and reveals multiple ways in which Innsbruck's mountainous location impacts its meteorology.
Fortunately for urban climatology and urban planners, many of the findings here are in accordance with previous
urban studies, for example:
• Energy partitioning in Innsbruck is similar to that in other city centres. The considerable thermal mass of the
urban surface stores a large amount of energy during the day and releases it at night.
• Near-surface stable conditions are rare as the release of stored heat and anthropogenic heat emissions maintain
a positive sensible heat flux all day and all year round.
• The low vegetation fraction around IAO keeps latent heat fluxes small and means there is very little opportunity
for water to be stored. Water supplied to the surface through precipitation thus has an impact (on the energy
balance and albedo) only for a short time (6-12 h) before it evaporates or is removed as run-off.
• In good agreement with the urban literature, the proportion of vegetation is a reasonable predictor of the
partitioning of energy between sensible and latent heat fluxes (in terms of summer daytime Bowen ratio) and
annual $CO_2$ exchange.
• The annual observed $CO_2$ flux of 5.1 kg C m$^{-2}$ y$^{-1}$ is dominated by anthropogenic emissions and is in reasonable
agreement with emissions estimated from a statistical inventory approach (5.0 kg C m$^{-2}$ y$^{-1}$), which suggests
traffic is the largest source of $CO_2$ during summer and building heating in winter. Future work to develop a
more advanced emissions model for Innsbruck will offer further insight.
However, Innsbruck's orographic setting and mountain weather affects near-surface conditions in multiple ways
and gives rise to specific features.
• The radiative fluxes are affected via orographic shading (incoming shortwave radiation is blocked by the terrain
so that local sunrise/sunset is later/earlier than over flat terrain). On convective days with clear skies over the
valley centre, cloud formation over the crests can further reduce solar radiation receipt during the late afternoon.
• Atmospheric transmissivity is related to the composition of the valley atmosphere which can be high in aerosols
(biogenic and anthropogenic pollutants), particularly during winter.
• The thermally driven valley-wind circulation in the Inn Valley gives rise to strong diurnal and seasonal cycles
in flow and turbulence. In Innsbruck the anabatic winds are stronger than the katabatic flow and the strength
and duration of the up-valley wind is greatest in spring and summer. In winter weak down-valley flow
dominates and the up-valley period is either very short or does not occur at all.
• These patterns complicate interpretation of local-scale measurements since the EC source area is biased
towards particular wind sectors for certain conditions. Fortunately, the relatively homogenous source area of
the IAO tower means the footprint composition does not change dramatically for different conditions. No clear
differences in energy partitioning related to source area characteristics could be identified but $CO_2$ fluxes are
considerably higher for the more densely built eastern sector with busier roads than for the western sector with
more vegetation and open water. As a result, the observed $CO_2$ fluxes likely underestimate the winter and
overestimate the summer emissions compared to the neighbourhood average.
• In spring and autumn south foehn events have a marked impact on conditions in Innsbruck. High wind speeds
and very large turbulent kinetic energies are observed which help to disperse urban pollutants (shown here by
very low $CO_2$ mixing ratios).
• The advection of warm air during foehn leads to negative sensible heat fluxes even in the urban environment
(and more often outside the urban area), especially in autumn and winter. The limited water availability appears
to restrict $Q_E$ in the urban environment compared to rural locations where enhanced $Q_E$ is observed. Although
$Q_H$ can be strongly negative during foehn these conditions are usually classified by the stability parameter
based on the Obukhov length as neutral (not stable) since wind speeds are high.
• The valley-wind circulation also seems to be responsible for reduced $Q_H$ and enhanced $Q_E$ during the
afternoons. This feature is most evident at rural sites with greater availability of water, where $Q_H$ can turn
negative long before sunset but also seems to reduce $Q_H$ in the afternoons at IAO.
Since the local- and mesoscale circulations that occur in mountainous regions also occur to some extent over less
complex terrain, as well as at great distances from the mountains, the results here are widely relevant, particularly





for densely populated coastal cities also subject to strong seasonal and diurnal variation in circulation patterns. For
the first time, this study describes the effects on urban surface-atmosphere exchange of a highly complex mountain
setting using the IAO site located on the floor of the Inn Valley; future work should consider urban surfaces on
sloping terrain.

**Data availability**

Data collected as part of the PIANO project are accessible via Zenodo (Gohm et al., 2021a; Gohm et al., 2021b;
Ward et al., 2021).

**Author contribution**

HCW and MWR designed the study. HCW conducted the analysis and prepared the manuscript with contributions
from all co-authors. All authors contributed to collection and processing of the various datasets involved.

**Competing interests**

TK is a member of the editorial board of *Atmospheric Chemistry and Physics*. The peer-review process was guided
by an independent editor and the authors have no other competing interests to declare.

**Acknowledgements**

This work was funded by the Austrian Science Fund (FWF) Lise Meitner programme (M2244-N32) and a
Research Stipend from Innsbruck University. Observations at IAO are possible thanks to Hochschulraum-
Strukturmittel funds provided by the Austrian Federal Ministry of Education, Science and Research, and the
European Commission Seventh Framework (ALP-AIR grant number 334084). Research at IAO is also supported
by FWF grants P30600-NBL and P33701-N. Part of the data collection and analysis were undertaken within the
framework of the PIANO project supported by the FWF and the Weiss Science Foundation (grant number P29746-
N32). We thank Florian Haidacher (Amt der Tiroler Landesregierung, Abteilung Mobilitätsplanung) for providing
official traffic data referenced in this study.

**Appendix A    Classification of conditions**

**A.1    Identification of flow regime**

To enable investigation of the impact different orographic flow regimes on surface-atmosphere exchange, the
clearest examples of thermally dominated (i.e. valley-wind) and dynamically dominated (i.e. foehn) events have
been identified. In reality there are few 'textbook' cases of thermally driven or dynamically driven events; most
days consist of a mixture of interacting processes across a wide range of scales. Hence there are very few days that
can be selected as examples of a purely thermally driven valley-wind circulation (Lehner et al., 2019). Similarly,
foehn events can vary considerably and the beginning and end of breakthrough periods are not easy to classify
(Mayr et al., 2018), nor distinguish from pre-foehn (when conditions are strongly affected by foehn flow aloft but
the foehn air does not reach the surface). Different flow regimes can give rise to similar temporal patterns in wind
speed and direction. For instance, breakthrough of foehn to the west of Innsbruck can appear similar to a thermally
driven up-valley flow, especially if the breakthrough occurs in the afternoon, appears as easterly flow and coincides
with increased wind speeds (e.g. as for FLUG in Figure 8). The complexity of the situation makes automated
classification extremely difficult. As it is not the intention here to develop algorithms that could be used more
generally to classify different conditions, manual classification was found to be the most useful approach to
facilitate understanding of the observations at IAO and examine the impact of these different conditions on near-
surface conditions.

**A.1.1    Valley-wind days**

Days with a transition from down-valley to up-valley flow and back again in both the Inn Valley and Wipp Valley,
with no obvious foehn or other flow type, are designated valley-wind days. The requirement for wind reversal also
in the Wipp Valley largely eliminates days with foehn flow in the Wipp Valley (for which a continuous southerly
wind is usually observed). However, days with weak synoptic forcing when a down-valley wind prevails in both





valleys (common in winter) are missed using this approach. Note that the requirement for a reversal to down-valley
flow again was met if it happened shortly after midnight in the Inn Valley, as the up-valley period can last until
late evening in summer (Figure 5). The results would not change significantly if the algorithm of Lehner et al.
(2019) had been used to select 'ideal' valley wind (i.e. synoptically undisturbed clear-sky) days, but the manual
classification includes many more days (and is not restricted to clear-sky conditions).
**A.1.2    Foehn events and pre-foehn conditions**
As a prerequisite for both foehn and pre-foehn conditions in the Inn Valley, south foehn had to be present in the
Wipp Valley (i.e. strong southerly winds at Steinach (S in Figure 1) and high foehn probabilities according to the
statistical mixture model of Plavcan et al. (2014)). Times when the foehn reached IAO were then judged from
visual inspection of the timeseries, largely based on closely matched potential temperatures at IAO and Steinach,
but changes in air temperature, relative humidity, wind speed and direction at IAO were also taken into account.
Unclear events, non-south foehn and very short possible breakthroughs were ignored. Hence not all foehn events
are captured by this approach, – but the majority of cases and the most clear-cut cases are. Comparison with the
statistical foehn diagnosis algorithms of Plavcan et al. (2014) revealed good agreement in terms of diurnal and
seasonal patterns, although the manual approach identifies slightly more cases in winter and night-time.
Disagreement between methods resulted from differences in availability of required input data, uncertainty about
the timing of the onset and cessation of foehn (e.g. foehn ceases but the warm air mass remains) and differences
between the approaches used (e.g. some of the statistical methods specify southerly wind directions but deflected
foehn can be from a range of directions in Innsbruck, see Section 5.4). The level of agreement and reasons for
differences between methods are similar to those discussed between the various algorithms presented in Plavcan
et al. (2014): accounting for differences in data availability 95% of the times identified as foehn by the manual
approach are also diagnosed as foehn by the statistical algorithm that is not restricted by wind direction. The results
would not change substantially if one of the foehn diagnoses of Plavcan et al. (2014) had been used instead of the
manual approach. For the pre-foehn conditions, again south foehn had to be present in the Wipp Valley but, in
contrast to a foehn breakthrough, the potential temperature at IAO is usually below the foehn temperature and
strong westerly winds ('pre-foehn westerlies') are often observed (Zängl, 2003).
**A.2    Identification of clear-sky days**
Clear-sky days in Innsbruck were identified using timeseries plots of 1-min incoming shortwave ($K_\downarrow$) and longwave
($L_\downarrow$) radiation at IAO plus visual inspection of webcam images to clarify ambiguous cases. Days with perfectly
smooth or almost perfectly smooth diurnal cycles of $K_\downarrow$ and $L_\downarrow$ were classified as clear-sky days at IAO (141 in
total). This manual approach was found to be more useful than using thresholds (e.g. of $K_\downarrow$ relative to $K_\downarrow$ at the top
of the atmosphere, $K_{\downarrow\,TOA}$) due to several complicating factors. These include (i) substantial variability in
atmospheric transmissivity even for cloud-free days (Figure 10a); (ii) short thunderstorms and associated cloud
cover which can develop in the late afternoon and do not dramatically affect daily $K_\downarrow/K_{\downarrow\,TOA}$; (iii) cumulus clouds
forming above the peaks and ridges (Section 6.1) but not above the valley centre and (iv) some very high $K_\downarrow/K_\downarrow$
$_{TOA}$ values despite mostly cloudy skies which occur when the sun shines through a gap in the clouds (observed $K_\downarrow$
is comprised of a large direct radiation component plus appreciable diffuse radiation).
**Appendix B    Estimation of anthropogenic heat flux and associated carbon dioxide emissions**
Anthropogenic heat flux and associated carbon dioxide emissions were estimated for the study area following a
conventional approach based on available statistics (e.g. Sailor and Lu, 2004) that combines contributions from
energy use within buildings, traffic and human metabolism. It is beyond the scope of this study to develop a
detailed emissions model for Innsbruck (this is planned in future); the aim here is to obtain a first-order estimation
to provide context for the observational analysis. For this section, all times refer to local time (UTC+1 or UTC+2
during daylight saving time).
The average metabolic energy release per person was assumed to be 175 W m$^{-2}$ cap$^{-1}$ when awake and 75 W m$^{-2}$
cap$^{-1}$ when asleep (Sailor and Lu, 2004). People are assumed to be awake 08:00-21:00 on working days and 09:00-
21:00 on non-working days and asleep 23:00-06:00 on working days and 23:00-07:00 on non-working days. The
population density on non-working days and during the night is 7000 km$^{-2}$ for the centre of Innsbruck (City
Population, 2018) and, based on the number of commuters (Statistik Austria, 2016), was estimated to increase by



25% during working hours to 8800 $km^{-2}$. Working hours are 08:00-16:00 and non-working hours are 18:00-06:00.
During the transition times, population density and metabolic energy release were linearly interpolated.
Multiplying these two quantities gave the anthropogenic heat flux due to human metabolism. For the corresponding
$CO_2$ release, emission factors of 280 μmol $s^{-1}$ $cap^{-1}$ and 120 μmol $s^{-1}$ $cap^{-1}$ were used for waking and sleeping
hours, respectively (Moriwaki and Kanda, 2004).
Domestic energy consumption for the study area was downscaled from the total energy consumption for the Tirol
region for 2015/2016 (Statistik Austria, 2017a) using population density. Energy consumption was available
separated into contributions from different energy sources (e.g. electricity, oil, wood, district heating and gas) and
for different purposes (classified as heating and non-heating purposes). All energy sources were assumed to
contribute to the anthropogenic heat flux, whilst the $CO_2$ emissions associated with electricity and district heating
were assumed to occur outside the study area and thus were not included in the $CO_2$ emissions estimated here.
Although wood burning is still common in smaller towns and villages in Tirol, it is no longer used much in
Innsbruck, so wood burning was apportioned to gas instead. Emission factors of 74.1 x $10^{-6}$ g $CO_2$ $J^{-1}$ for oil and
56.1 x $10^{-6}$ g $CO_2$ $J^{-1}$ for gas were used (IPCC, 2006). Daily energy use for non-heating purposes was assumed
constant, whilst daily energy use for heating purposes was assumed to scale with heating degree days (Sailor and
Vasireddy, 2006): an increase in energy use of 0.45 W $m^{-2}$ $K^{-1}$ was obtained using a base temperature of 18.3 °C
(Sailor et al., 2015), which appears to be a reasonable threshold for Innsbruck based on Figure 15c. The daily
values were then downscaled to 30 min using standard profiles of building energy use for working days and non-
working days in Austria (Ghaemi and Brauner, 2009). Non-domestic building energy use is estimated as a
proportion (0.42) of domestic building energy use, given that 26%/11% of the total building energy use in Europe
is domestic/non-domestic (Pérez-Lombard et al., 2008). Non-domestic building energy use was assumed to be
70% and 50% of the working-day value for Saturdays and for Sundays and holidays, respectively. Non-domestic
energy use profiles from Hamilton et al. (2009) were used to downscale daily values to 30 min.
The total number of kilometres driven by passenger cars in Tirol in 2015/2016 (Statistik Austria, 2017b) was
scaled by population density for the centre of Innsbruck to estimate the weight of traffic in the study area
(contributions from motorcycles, public transport or goods transport are neglected in this approach). Hourly traffic
count data for 7 stations around Innsbruck for Jan 2018-Jun 2020 (provided by Amt der Tiroler Landesregierung,
Abteilung Verkehrsplanung) were used to derive average traffic rates and median diurnal cycles for four groups:
Monday-Thursdays; Fridays; Saturdays; and Sundays and holidays. Traffic rates on Fridays, Saturdays, and
Sundays and holidays were about 105%, 77%, and 55% of traffic rates on Monday-Thursdays, respectively.
Monthly traffic counts for the study period for the 7 stations around Innsbruck (Land Tirol, 2020) were used to
account for monthly and interannual variation in traffic rates across the whole study period (e.g. due to Coronavirus
restrictions). From the average traffic weight (scaled according to month, type of day and time of day), the heat
flux was calculated assuming an emission factor of 3.97 MJ $km^{-1}$ $veh^{-1}$ (Sailor and Lu, 2004). For $CO_2$ an emission
factor of 0.17 kg $CO_2$ $km^{-1}$ $veh^{-1}$ was used (Statistik Austria, 2017b). Variation of emission factor with speed and
type of vehicles was neglected as specific information was not available.
**Appendix C      Calculation of net storage heat flux**
The Objective Hysteresis Model (OHM) (Grimmond et al., 1991) estimates the net storage heat flux $\Delta Q_S$ from the
net radiation $Q^*$:
$$\Delta Q_S = \sum_i f_i \left[ a_{1i} Q^* + a_{2i} \frac{\partial Q^*}{\partial t} + a_{3i} \right], \qquad\qquad\qquad (C1)$$
where $f$ is the proportion of each surface cover type $i$, $a_{1,2,3}$ are coefficients for each surface cover type and $t$ is
time. The coefficients were taken from the literature (Table C 1) and resulted in bulk values for 500 m around IAO
of 0.475, 0.287 and -33.1 for $a_1$, $a_2$ and $a_3$, respectively. Note that deriving bulk coefficients using observed $Q^*$
and $\Delta Q_{S\_RES}$ showed little seasonal variation at IAO in contrast to other studies (Anandakumar, 1999; Ward et al.,
990 2016).

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

| Surface type | $a_1$ | $a_2$ [h$^{-1}$] | $a_3$ [W m$^{-2}$] | Source |
|---|---|---|---|---|
| Buildings | 0.477 | 0.337 | -33.9 | Average of Yap (1973), Taesler (1980) and Yoshida et al. (1990); Yoshida et al. (1991) |
| Paved areas | 0.665 | 0.243 | -42.8 | Average of Narita et al. (1984), Doll et al. (1985) and Asaeda and Ca (1993) for asphalt and concrete |
| Road | 0.500 | 0.275 | -31.5 | Average of Narita et al. (1984) and Asaeda and Ca (1993) for asphalt |
| Water | 0.500 | 0.210 | -39.1 | Souch et al. (1998) |
| Short vegetation | 0.320 | 0.540 | -27.4 | Short grass values from Doll et al. (1985) |
| Trees | 0.110 | 0.110 | -12.3 | Mixed forest values from McCaughey (1985) |
| Other | 0.355 | 0.333 | -35.3 | Average of Fuchs and Hadas (1972), Novak (1981) and Asaeda and Ca (1993) for bare soil |

**Table C 1: Coefficients for each surface type used in the Objective Hysteresis Model.**

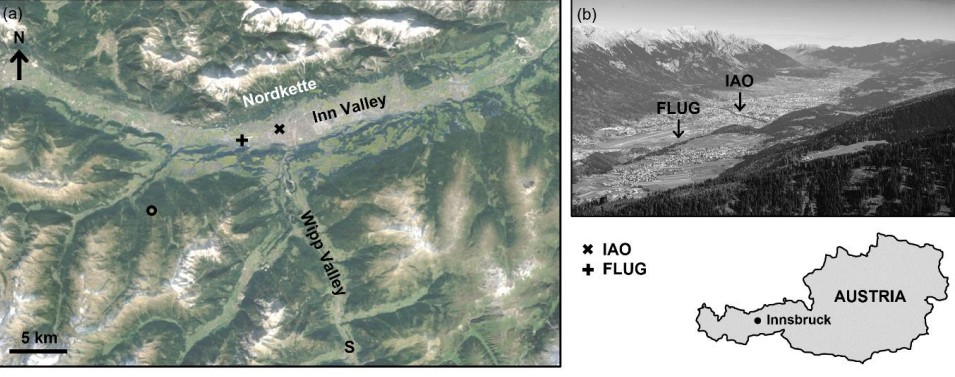

**Figure 1: (a) Location of the Innsbruck Atmospheric Observatory (IAO) and airport (FLUG) sites in the**
**Inn Valley and Steinach (S) in the Wipp Valley (aerial imagery from © Google Earth). (b) Photograph taken**
**from the position marked with an open circle in (a) looking eastwards along the Inn Valley over Innsbruck**
**airport and the city of Innsbruck. The location of Innsbruck within Austria is shown (bottom right).**



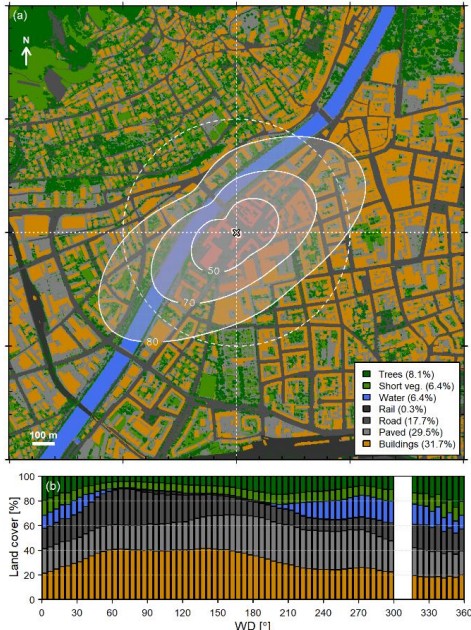

**Figure 2: (a) Composite source area at IAO for the study period (01 May 2017-30 Apr 2021) superimposed on land cover (derived from various spatial datasets from Land Tirol, data.tirol.gv.at). Contours indicate the region comprising 50, 70 and 80% of the source area and the circle indicates a distance of 500 m from the flux tower. (b) Average land cover composition by wind direction (no data are available for wind directions of 309 ± 10°, Section 2.3). The aggregated source area composition for the study period is given in the legend.**



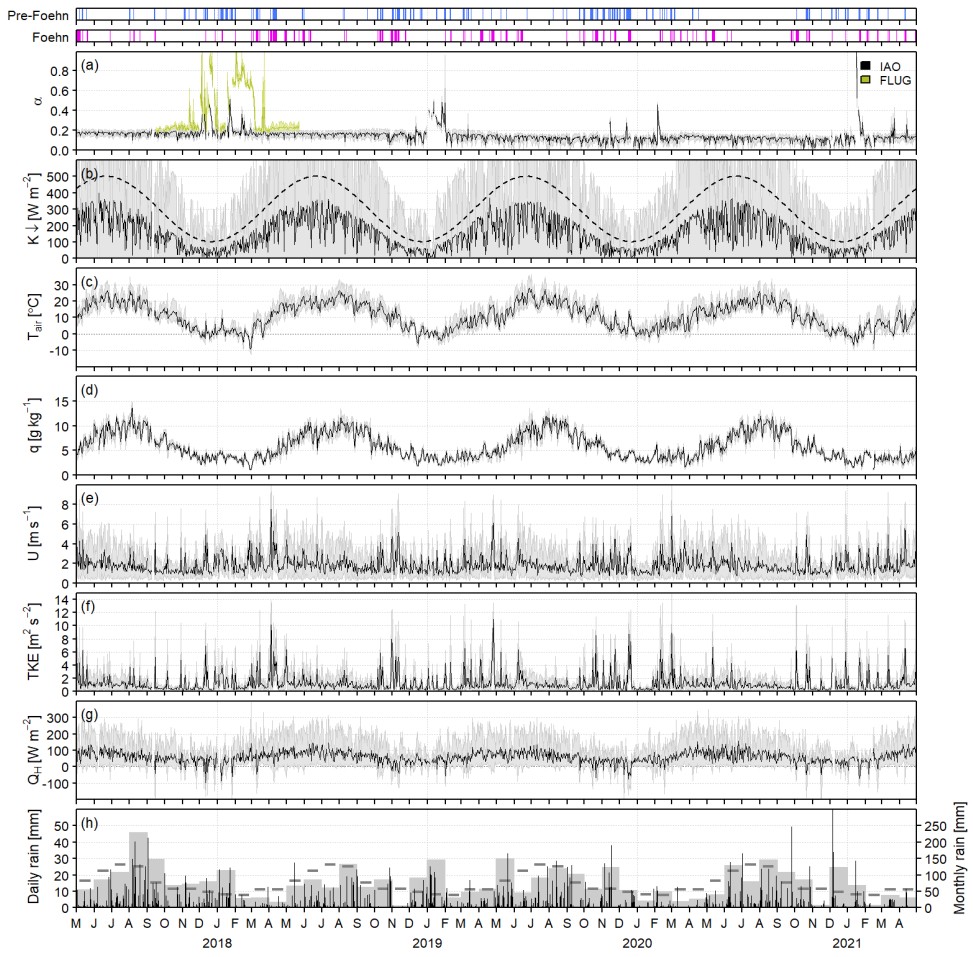

1392

**Figure 3: Time series of daily mean values (shading indicates 10-90[th] percentiles) of (a) albedo (b) incoming shortwave radiation (c) air temperature (d) specific humidity, $q$, (e) wind speed, (f) turbulent kinetic energy and (g) sensible heat flux; and (h) daily and monthly rainfall for IAO. In (h) the thick horizontal bars indicate the 1981-2010 normal monthly rainfall (ZAMG, 2021). The albedo for site FLUG is also shown in (a). In (b) the dashed line indicates top-of-atmosphere irradiance. The top panels show the occurrence of foehn and pre-foehn conditions (see Appendix A for details).**



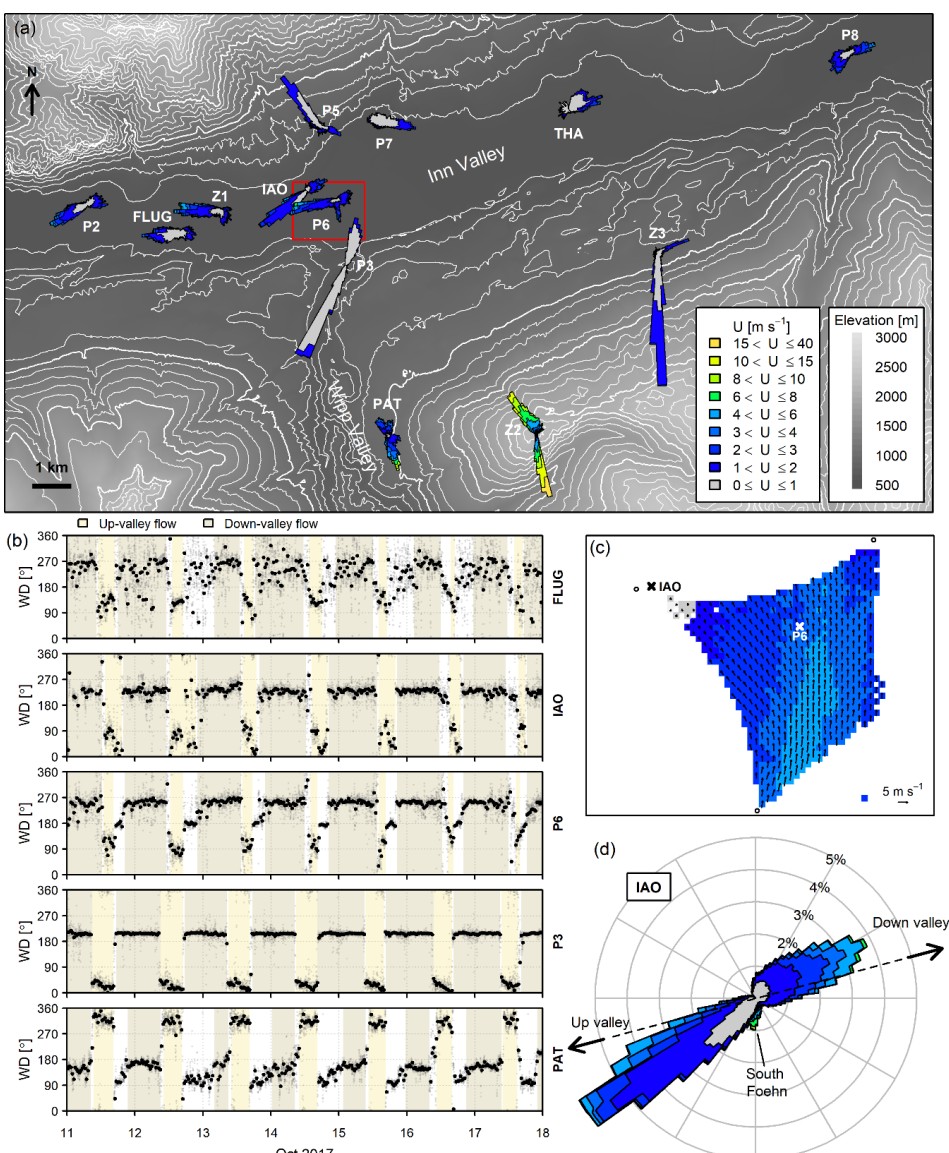

1399

**Figure 4: (a) Wind roses for stations in and around Innsbruck during Autumn 2017 (all available 30-min data for September-November 2017) overlaid on a digital elevation map (data source: Amt der Tiroler Landesregierung, Abteilung Geoinformation) with thin/thick contours at 100-m/500-m intervals. (b) Timeseries of wind direction for selected sites for the mostly clear-sky period 11-18 October 2017 with up-valley and down-valley flow shaded. 30-min data are shown in black and 1-min data in grey. (c) Horizontal wind field (approx. 60 m above ground) at 17:50-18:00 CET on 16 October 2017 derived from three Doppler wind lidars (circles) performing coplanar scans (see Haid et al. (2020) for details). Colours/arrows indicate wind speed/wind direction; points outside the coplanar field of view are left white. The area shown in (c) corresponds to the red box in (a) and is about 1.8 km x 1.5 km. (d) Wind rose for IAO during the study period (01 May 2017-30 Apr 2021). The black line indicates the approximate orientation of the Inn Valley axis at IAO with the up- and down-valley directions marked. The colour scale for wind speed in (c) and (d) is the same as in (a).**

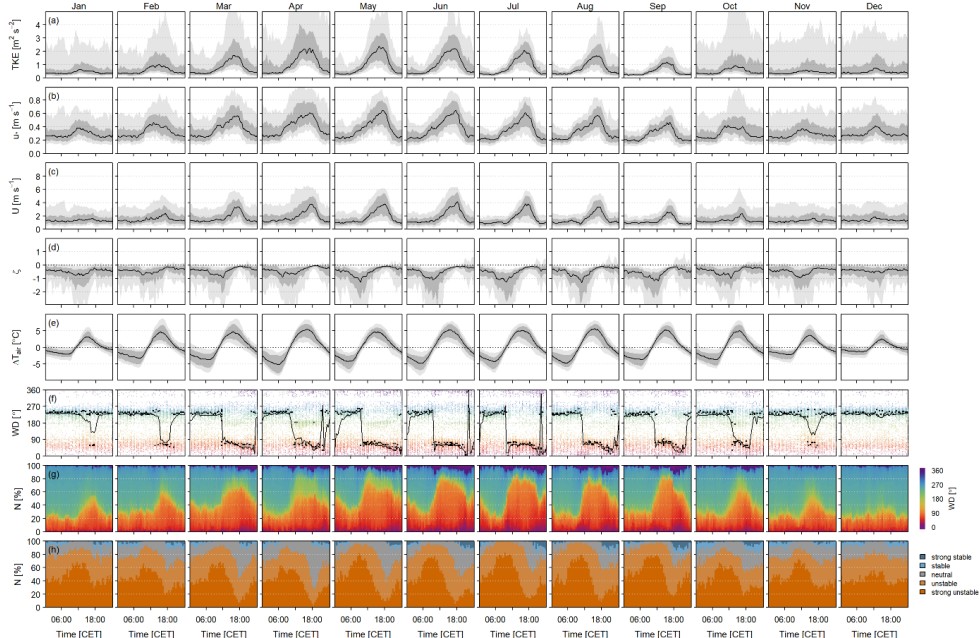

1412

**Figure 5: Monthly median diurnal cycles (black lines), interquartile ranges (dark shading) and 10-90th percentiles (light shading) of (a) turbulent kinetic energy, (b) friction velocity, (c) wind speed, (d) dynamic stability and (e) difference between 30-min and daily mean air temperature, $\Delta T_{air}$. (f) Monthly median diurnal cycles (black lines), modal values (black points) and individual 30-min wind directions (coloured points). Normalised frequency distributions of (g) wind direction and (h) stability separated by month and by time of day. Stability classes are strongly unstable ($\zeta \le$ -0.5), unstable (-0.5 $< \zeta \le$ -0.1), neutral (-0.1 $< \zeta \le$ 0.1), stable (0.1 $< \zeta \le$ 0.5) and strongly stable ($\zeta >$ 0.5).**

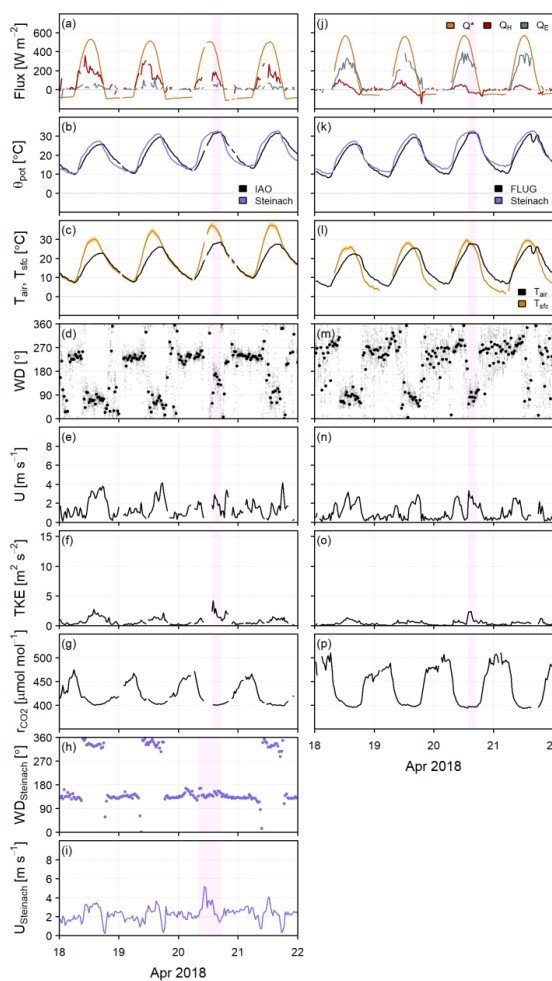

**Figure 6: Timeseries of (a, j) net radiation, sensible heat flux and latent heat flux (b, k) potential temperature, $\theta_{pot}$, (c, l) air temperature and surface temperature derived from outgoing longwave radiation assuming an emissivity of 0.95 (shading indicates the uncertainty for emissivities 0.90-1.00), (d, m) wind direction, (e, n) wind speed, (f, o) turbulent kinetic energy and (g, p) $CO_2$ mixing ratio for the clear-sky period 18-22 April 2018 at (a-g) IAO and (j-p) FLUG. Wind direction (h), wind speed (i) and potential temperature (b, k) at Steinach in the Wipp Valley are also shown. All data are at 30-min resolution; 1-min wind direction is additionally shown at IAO and FLUG (small grey points in (d) and (m)). Shading indicates times with foehn at each site (see Appendix A for details).**





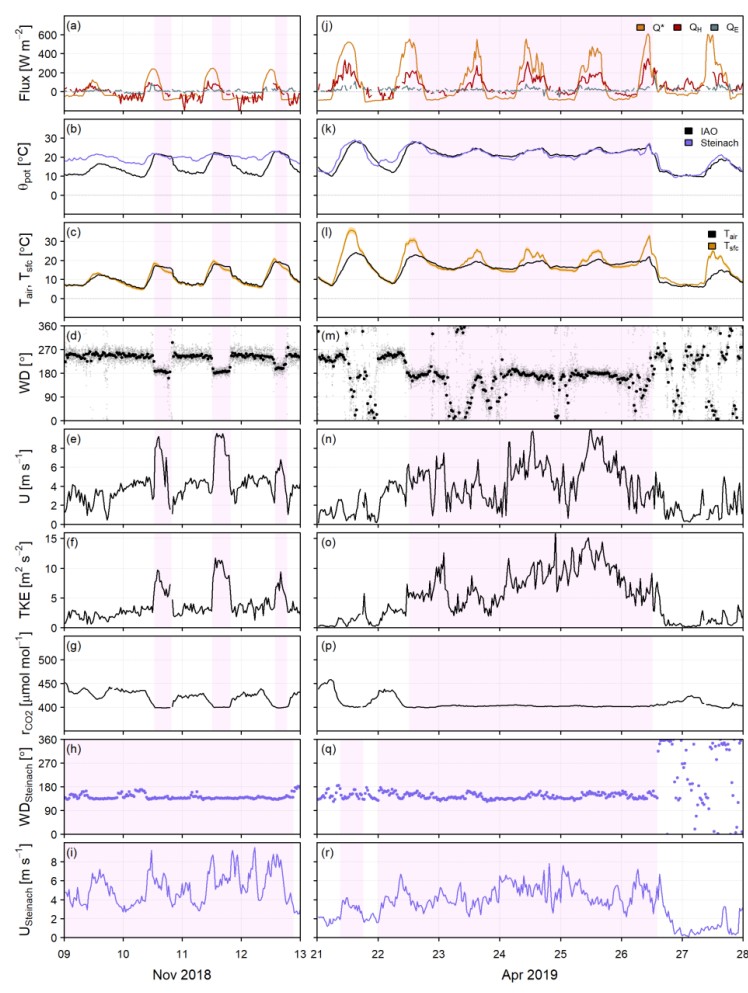

1429

**Figure 7: As Figure 6a-i for the periods (a-i) 09-13 November 2018 and (j-r) 21-28 April 2019 affected by foehn. Data are shown for (a-g, j-p) IAO and (h-i, q-r) Steinach.**

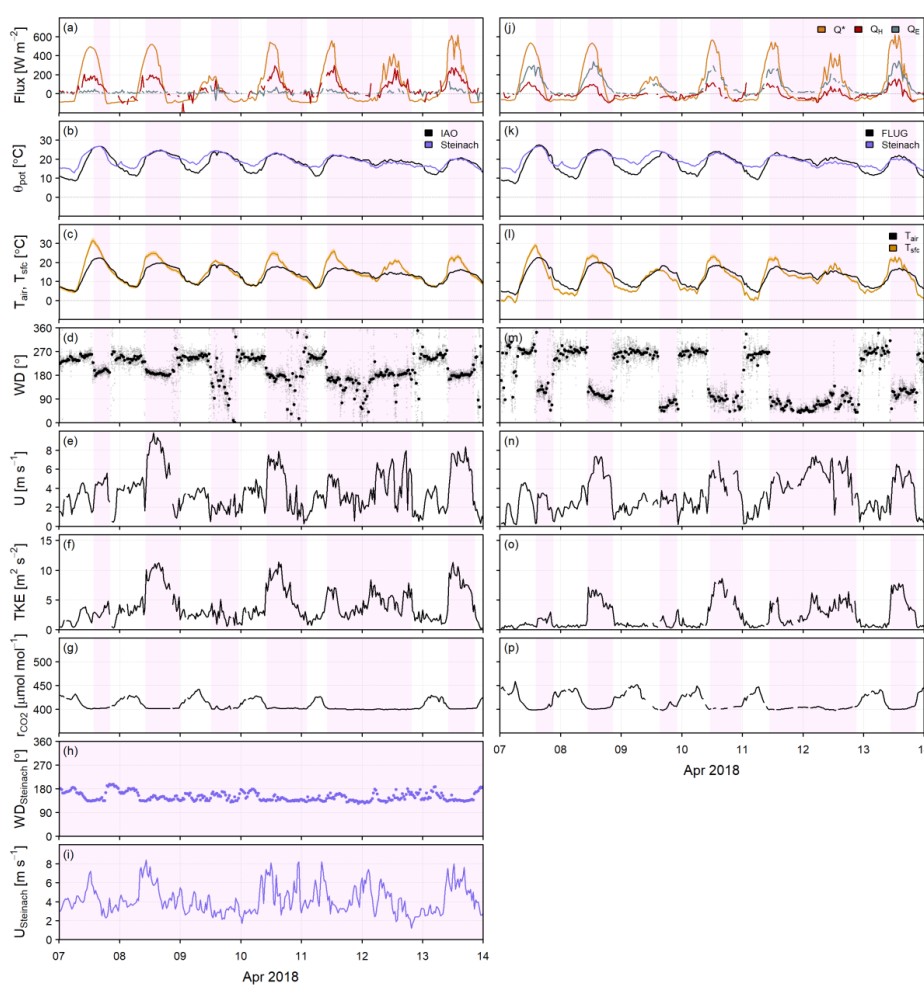

1432

**Figure 8: As Figure 6 for the period 07-14 April 2018 at (a-g) IAO, (j-p) FLUG and (h-i) Steinach.**





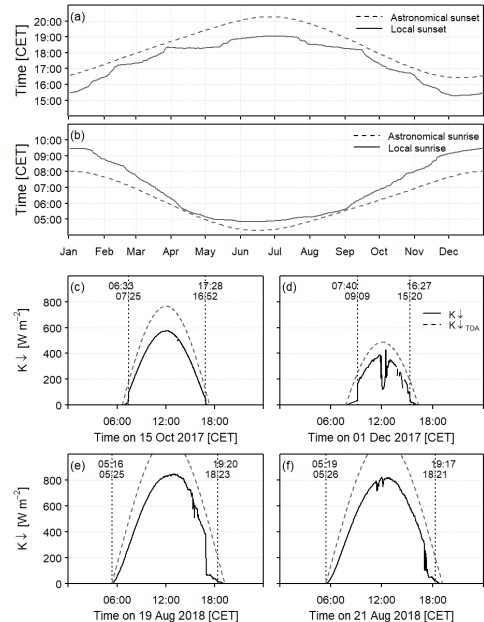

1434

**Figure 9:** Time of local and astronomical (a) sunset and (b) sunrise at IAO. Observed incoming shortwave radiation (1-min data) for example days illustrating the effect of (c, d) orographic shading and (e, f) orographic shading plus afternoon cloud cover around the surrounding peaks. In (c-f) local sunrise and sunset are marked by dotted vertical lines and the times of local and astronomical sunrise and sunset are shown. The incoming shortwave radiation at the top of the atmosphere $K_{\downarrow TOA}$ is also shown (see text for details).

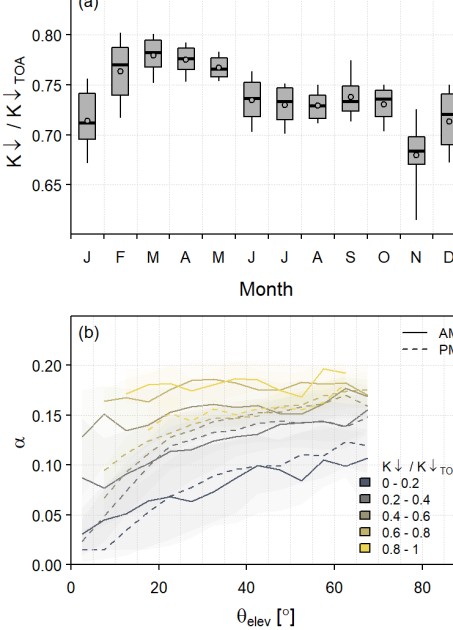

1441





**Figure 10: (a) Monthly boxplots of the midday (11:00-15:00 CET) clearness index $K_\downarrow / K_{\downarrow\,TOA}$ at IAO for clear-sky days. Boxes indicate the interquartile range, whiskers the 10-90th percentiles and the median and mean are shown by horizontal bars and points, respectively. (b) Albedo ($K_\downarrow > 5$ W m$^{-2}$, times with snow cover have been removed) at IAO versus elevation angle separated by clearness index and into morning (AM) and afternoon (PM) periods. Lines indicate binned median values and shading the interquartile range.**

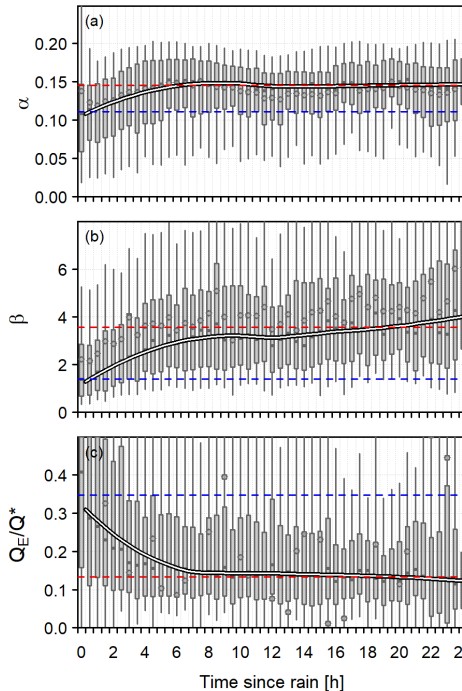

**Figure 11: Boxplots of (a) albedo, (b) Bowen ratio and (c) the ratio of the latent heat flux to net radiation binned by time since (> 0 mm) rainfall for daytime ($K_\downarrow > 5$ W m$^{-2}$) data only at IAO. The thick line is a loess curve through the median values; the red dashed line indicates the mean of these values between 12 and 24 hours since rain (i.e. reasonably dry conditions) and the blue dashed line for the first two boxes (i.e. wet conditions less than one hour since rainfall).**



1453

**Figure 12: (a-k) Monthly median diurnal cycles (lines), interquartile ranges (dark shading) and 10-90th**
**percentiles (light shading) of radiative fluxes, energy balance terms and carbon dioxide fluxes: (a) incoming**
**shortwave radiation, (b) outgoing shortwave radiation, (c) incoming longwave radiation, (d) outgoing**
**longwave radiation, (e) net all-wave radiation, (f) anthropogenic heat flux, (g) sensible heat flux, (h) latent**
**heat flux, (i) storage heat flux, (j) Bowen ratio and (k) carbon dioxide flux. All available data for IAO and**
**FLUG are shown in black and green, respectively. In (i) the net storage heat flux ($\Delta Q_S$) estimated using the**
**Objective Hysteresis Model (OHM) and estimated as the energy balance residual (RES) is shown for IAO**
**and the ground heat flux ($Q_G$) is shown for FLUG. (l-v) Barplots of daily mean fluxes at IAO separated by**
**month and by year (colours). In (t) the storage heat flux is estimated using OHM. Note the different y-axis**
**limits.**

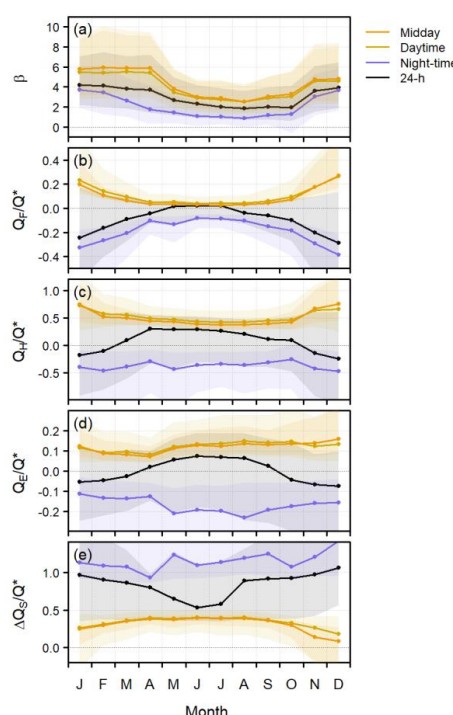

1464

**Figure 13: Energy partitioning at IAO for different subsets: midday (11:00-15:00 CET), daytime ($K_\downarrow > 5$ W m⁻²), night-time ($K_\downarrow \leq 5$ W m⁻²) and 24-h. Lines indicate monthly median values and shading the interquartile range for (a) Bowen ratio and for (b) anthropogenic heat flux, (c) sensible heat flux, (d) latent heat flux and (e) net storage heat flux (calculated using OHM) normalised by net radiation.**

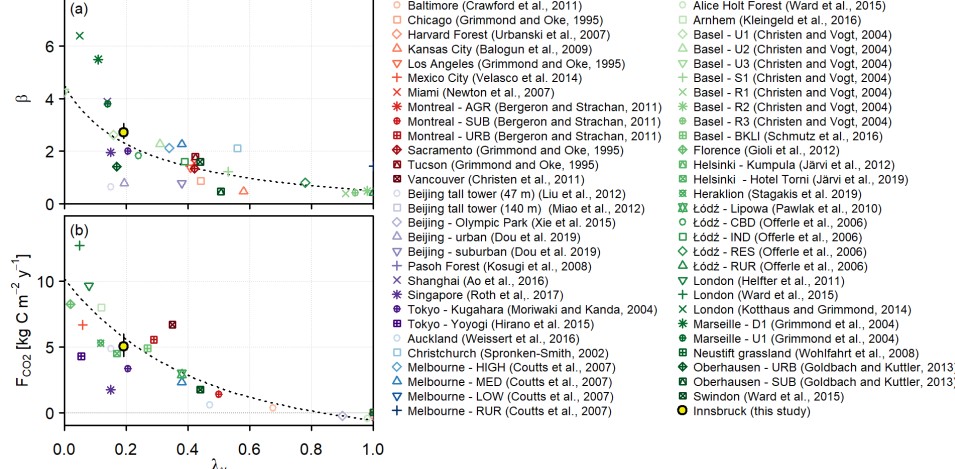

1469

**Figure 14: (a) Daytime (or midday if daytime is not given in the corresponding publication) Bowen ratio during summer and (b) annual net carbon flux versus vegetation fraction $\lambda_v$ for IAO and for various sites in the literature (see legend for references). Error bars for IAO indicate the spread of (a) daytime summertime values for the different years and (b) annual totals over the four twelve-month periods of the**



**dataset. The dotted lines in (a) are Equation 3 from Christen and Vogt (2004) and in (b) from Nordbo et al.**
**(2012).**

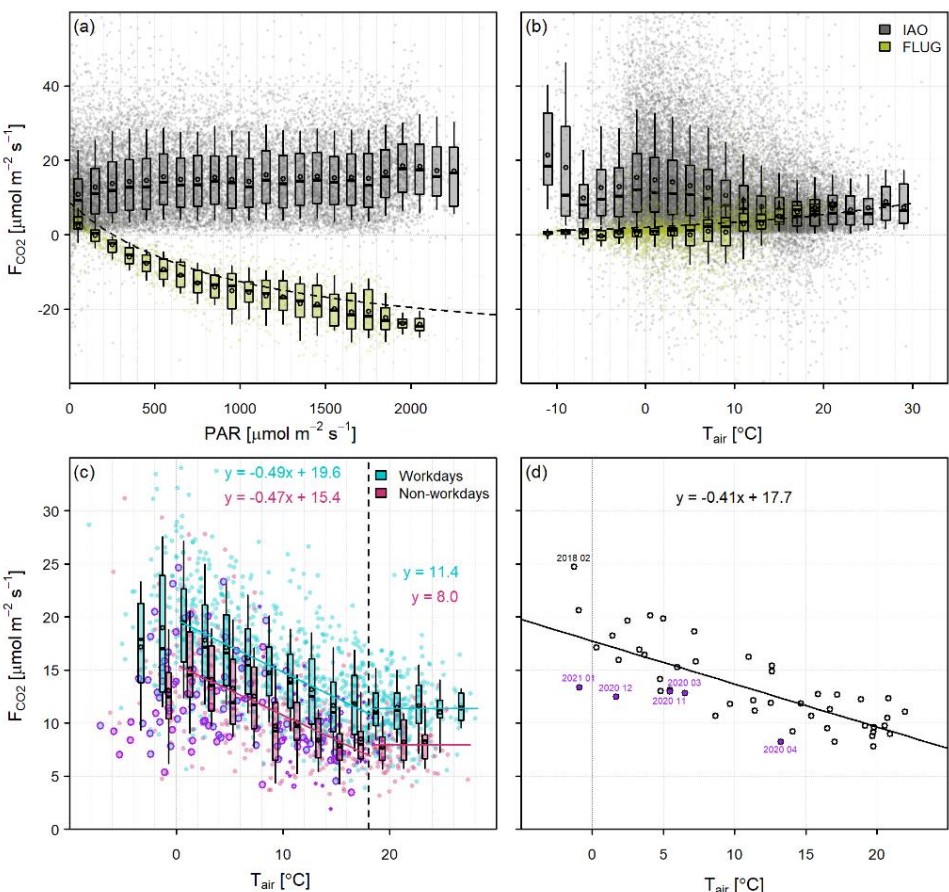


**Figure 15: (a) 30-min daytime ($K_\downarrow > 5$ W m$^{-2}$) observed CO$_2$ fluxes versus photosynthetically active radiation**
**for all available data during the growing season (April-September, inclusive, but note that FLUG data are**
**only available for 15 September 2017-22 May 2018); and (b) 30-min night-time ($K_\downarrow \leq 5$ W m$^{-2}$) observed CO$_2$**
**fluxes versus air temperature for the urban (IAO) and grassland (FLUG) sites. The dashed lines show (a)**
**the light-response curve and (b) soil respiration rate for a grassland site in the nearby Stubai Valley**
**(Wohlfahrt et al., 2005; Li et al., 2008). (c) Daily mean CO$_2$ flux versus daily mean air temperature separated**
**into working and non-working days (daily values have been gap-filled using monthly median diurnal cycles**
**for working and non-working days). Points outlined in purple occurred during Coronavirus restrictions.**
**The vertical dashed line marks a base temperature of 18 °C above which $F_{CO2}$ does not decrease with**
**temperature. (d) Average monthly observed CO$_2$ fluxes versus average monthly air temperature. Purple**
**points indicate months with the strictest Coronavirus restrictions. In (a-c) boxes indicate the interquartile**
**range, whiskers the 10$^{th}$-90$^{th}$ percentile, horizontal bars the median and points the mean. In (c-d) solid lines**
**are linear regressions with the equations given.**

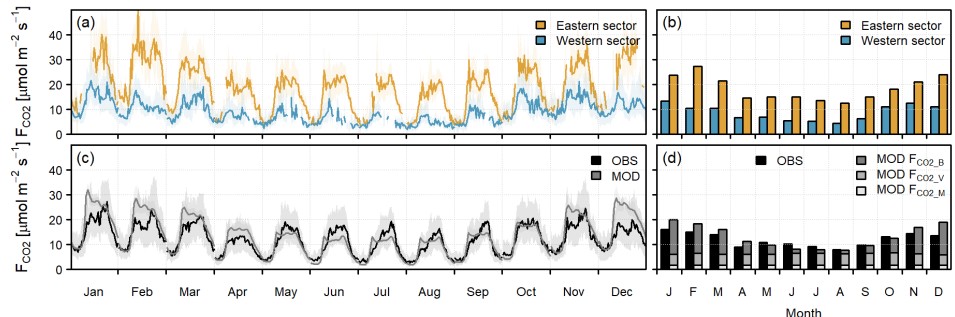


**Figure 16: (a, c) Monthly median diurnal cycles (shading indicates interquartile range) and (b, d) the corresponding daily mean fluxes by month for (a-b) observed carbon dioxide fluxes separated into east (60-120°) and west (210-270°) wind sectors and (c-d) observed and modelled carbon dioxide fluxes. In (d) the modelled emissions are separated into contributions from building heating ($F_{CO2\_B}$), traffic ($F_{CO2\_v}$) and human metabolism ($F_{CO2\_M}$).**



1496

**Figure 17: Impact of different flow regimes (colours) on (a, k, u) turbulent kinetic energy, (b, l, v) wind speed, (c, m, w) gust speed, (d, n, x) air temperature, (e, o, y) difference between surface and air temperature, (f, p, z) sensible heat flux, (g, q, A) sensible heat flux ratio, (h, r ,B) latent heat flux, (I ,s, C) latent heat flux ratio and (j, t, D) $CO_2$ mixing ratio at IAO. Boxplots are shown for all data together (a-j) and separated by month (k-t); boxes indicate the interquartile range, whiskers the 10-90th percentiles and the median and mean are shown by horizontal bars and points, respectively. Median diurnal cycles (lines), interquartile ranges (dark shading) and 10-90th percentiles (light shading) are separated by season (u-D). The category 'All' refers to the whole dataset (i.e. includes the other categories) and the proportion of the study period classified as each of the other categories is given above (a). Data are plotted if more than 5 data points are present.**