# Peer review of "Energy and mass exchange at an urban site in mountainous"

_Atmospheric Chemistry and Physics, 2021_

## Author Response (AR1)

**Response to Reviewers**

We thank the reviewers for their extremely positive assessment of the manuscript, for their careful read, interest in the results and suggestions for improvement. Individual responses are provided below.

Since both reviewers recommended the manuscript be published as is, very few changes have been made. Besides the minor additions described below, please note that the flux data for 1 Jan 2021-30 Apr 2021 have been reprocessed using spectral corrections which are now based on data for the whole of 2021 and not only Jan-Apr 2021 as was previously the case. This has a negligible impact – the only discernible (but still very minor) change is to the regression line in Figure 15d, which now has a gradient of -0.40 (previously -0.41). We state this only in the interest of transparency; the results are unchanged.

**Comments of Anonymous Referee #1**

Understanding singularities of the urban climate is becoming increasingly important in the light of contemporary climate change. Cities are the main point sources of the greenhouse gases and, on the other hand, areas that are vulnerable to climate change. Modifications of the elements of local climate by urbanization (e.g. urban heat island) have been intensively studied and are well recognized for the "ideal" case, i.e. a city surrounded by flat homogenous rural areas, especially in mid-latitudes. Similar modifications in the case of mountain or coastal cities will remain an open issue due to the possible interactions of various local factors. The presented manuscript fits perfectly into this research needs.

Moreover, the study is not limited to the classical meteorological elements, but also includes analysis of turbulent exchange mass, energy and momentum between the city's surface and atmosphere. Measurements of turbulent fluxes are extremely difficult in urban areas, which makes such studies unique.

For these reasons, I am absolutely convinced that the manuscript should be published in the ACP. It is very well prepared, following the standards of scientific articles. The flow of argumentation is clear, and I have found nothing that needs to be corrected, expanded or shortened. Thus, I suggest to publish manuscript as it is, with one technical correction: unit of parameter a2 in Table C1 should be [h] not [1/h].

Thank you for spotting this – the units are now correctly given as [h].

The following are issues that may be considered, but do not necessarily need to be corrected in the manuscript.

In ln. 251-252 authors state that "the reason for this 20-30° difference between the valley axis and the main wind directions is not clear" – I agree, but one can speculate, that it is related to the friction effect at anemometer height. The upper (in boundary layer) wind might probably be parallel to the valley axis, but at the height of the anemometer the friction reduces its speed and causes a left turn, like in an Ekman spiral. Of course, the valley wind is not geostrophic, but like any moving object, it is subject to Coriolis force, counterbalanced by other forces. When the speed is reduced by friction, the Coriolis force weakens and surface wind turns to the left.

We agree that this could be a contributing factor, however wind data from a nearby weather station (at the other side of the rooftop) at a similar height shows a slightly different wind direction distribution again. We therefore believe these differences in near-surface winds are more likely related to the effects of the urban surface or very local terrain features. Detailed modelling studies and further measurements are planned to investigate this further so we decided not to speculate at this stage.

In ln. 427-428 it is written: "on mostly clear-sky days cumulus clouds often form above the mountain peaks during the afternoons, making sunset appear even earlier since the sun is blocked by clouds before it is blocked by terrain" – the phenomenon is correctly described, but is it really an earlier "sunset"? In this way, any obscuring of the sun's disc by clouds could be called a "sunset". Fortunately, a more accurate wording was used in the summary (ln. 826): "cloud formation over the crests can further reduce solar radiation".

Thank you for picking this up - we agree the wording was inaccurate. The text now reads: 'on mostly clear-sky days cumulus clouds often form above the mountain peaks during the afternoons, causing an even earlier drop in solar radiation since the sun is blocked by clouds before it is blocked by terrain'.

I like analysis of CO2 flux in relation to source area (ln. 692-716). But, in the light of these results, it may be considered what could be the best method to calculate the annual exchange of CO2 in the vicinity of the measuring point. Here it is calculated as an average of the gap-filled CO2 flux dataset – standard approach. But perhaps it would be more realistic to divide the data into two data sets, fill the gaps and the next average both values. It is almost done in ln. 725-726 and result is pretty the same as in ln. 718.

Yes, averaging the totals for the eastern and western sectors separately also results in an average of 5.1 kg C m$^{-2}$ y$^{-1}$. The sentence has been extended to read: 'Considering the eastern and western sectors separately at IAO gives annual totals of 7.0 and 3.3 kg C m$^{-2}$ y$^{-1}$, respectively (which rather fortuitously average to give 5.1 kg C m$^{-2}$ y$^{-1}$, in agreement with the measured total given above).'

In conclusions (ln. 848-850) the reduced QH and enhanced QE during the afternoons at IAO is attributed to the valley-wind circulation with mechanism similar as at rural site, but is it not possible that this is a similar effect of changes in the source area as in the case of the diurnal asymmetry of the CO2 flux in the summer?

This is an interesting consideration. It is possible that the changing source area has a small influence on the temporal signatures of observed QH and QE. While CO2 fluxes show a marked systematic difference for eastern and western wind sectors which can be related to the source area characteristics, no major differences and no clear trends in energy partitioning were seen. The situation is, however, more complex for the energy partitioning – since the valley-wind circulation is a thermally-driven circulation, easterly up-valley winds occur more often under fair weather conditions. The expected impact of changing source area characteristics would, however, actually produce the opposite effect to that seen: the up-valley winds occur on warm, dry summer afternoons (high QH expected) and correspond to an easterly source area towards the city centre which is more built-up with less vegetation and less water compared to the western sector (again high QH expected). These would both mean that QH would be higher during summer afternoons – the opposite to what is observed. We are therefore confident that the effect is not due to changing source area characteristics. Furthermore, since a consistent trend is observed (but even more clearly) at the rural site (which is more homogeneous than the city) and at several other sites in the Inn Valley (see references in L620-1), we believe the valley-wind circulation and difference between air and surface temperatures (seen in Fig 6) is the main factor responsible for reduced QH/enhanced QE during the afternoons.

**Comments of Anonymous Referee #2**

This paper by Ward et al. presents an analysis of 4 years of micrometeorological data collected in Innsbruck, Austria, a small alpine city located at the bottom of Inn Valley and surrounded by complex topography. The analysis is mostly structured around radiation and energy budgets, with a particular focus on the impact of different meteorological patterns (valley-wind days vs foehn events vs pre-foehn events). If it is obvious that such a dataset is unprecedented, there is a great challenge to propose an interesting synthesis article. Well, the challenge was met! The article proposed by Ward is excellent from all points of view! The introduction is great, the objectives well defined and the methodology is clear. All the results are discussed exhaustively and put in perspective with numerous studies in the literature. The plots are flawless. The discussion is of high quality and the conclusions are relevant to the community and reflect well the results of the analysis. I have reviewed over 50 articles in my career and this is the first time I recommend accepting a manuscript as is (and without hesitation even)!

Among the points I liked most about the paper, the mixing ratio of CO2 that becomes completely flat during foehn events, the quantification of energy storage in the urban canopy and anthropogenic energy sources (heating, etc.), the fact that stable conditions are very rare, even in winter, and the link between atmospheric boundary layer properties and atmospheric transmissivity.

I take my hat off to all the co-authors of this excellent paper which fully deserves to be published in ACP.

Thank you very much for this exceptionally positive review! We worked hard to produce a comprehensive and worthwhile publication which addresses the complexities of this urban site in mountainous terrain, so we are very happy to hear that our efforts paid off.